# Abstract Rendering: Certified Rendering Under 3D Semantic Uncertainty

**Chenxi Ji**, **Yangge Li**, **Xiangru Zhong**, **Huan Zhang, Sayan Mitra**
University of Illinois Urbana-Champaign
`{chenxij2, li213, xiangru4, huanz, mitras}@illinois.edu`

## Abstract

Rendering produces 2D images from 3D scene representations, yet how continuous variations in camera pose and scenes influence these images—and, consequently, downstream visual models—remains underexplored. We introduce **abstract rendering**, a framework that computes provable bounds on all images rendered under continuously varying camera poses and scenes. The resulting abstract image, expressed as a set of constraints over the image matrix, enables rigorous uncertainty propagation through downstream neural networks and thereby supports certification of model behavior under realistic 3D semantic perturbations, far beyond traditional pixel-level noise models. Our approach propagates camera pose uncertainty through each rendering step using efficient piecewise linear bounds, including custom abstractions for three rendering-specific operations—matrix inversion, sorting-based aggregation, and cumulative product summation—not supported by standard tools. Our implementation, ABSTRACTRENDER, targets two state-of-the-art photorealistic scene representations—3D Gaussian Splats and Neural Radiance Fields (NeRF)—and scales to complex scenes with up to 1M Gaussians. Our computed abstract images achieve up to $3\%$ over-approximation error compared to sampling results (baseline). Through experiments on classification (ResNet), object detection (YOLO), and pose estimation (GATENet) tasks, we demonstrate that abstract rendering enables formal certification of downstream models under realistic 3D variations—an essential step toward safety-critical vision systems.

## 1 Introduction

Rendering produces 2D images from 3D scenes and underpins visual computing. Two prominent neural scene representations are Gaussian Splats [1] and Neural Radiance Fields (NeRFs) [2]. Gaussian Splats represent a scene as a collection of 3D Gaussians whose colors are blended after projection onto the image plane, whereas NeRFs encode scenes as neural networks mapping 3D positions and viewing directions to color and density, rendered via ray casting and volumetric integration. These methods exemplify rasterization- and ray-casting-based paradigms that achieve photorealistic reconstruction and novel-view synthesis [3]. However, rigorous analysis of how variations in camera pose or scene geometry affect the rendered outputs—and consequently the predictions of downstream models such as classifiers, object detectors, and pose estimators—remains limited, leaving the robustness of vision systems under realistic 3D perturbations largely unexplored.

Although formal verification has been extensively developed for standalone neural networks [4, 5], verifying rendering pipelines poses a fundamentally different challenge. A recent work [6] considers a version of this problem for simpler mesh-based scenes, but a general treatment remains open. Formal verification of rendering seeks to compute the set of *all* possible images generated under continuous variations in camera pose and scene. When coupled with downstream perception models this enables guarantees like "no misclassification occurs during camera panning within a specified range" or

---

*Equal contribution.

39th Conference on Neural Information Processing Systems (NeurIPS 2025).

"the relative pose error remains bounded within tolerance across viewpoints." Such guarantees are crucial for safety-critical applications [7] including aircraft auto-landing [8] and formation-flight[9]. Verifying rendering is challenging because ray-casting and rasterization involve neural components, volumetric integration, and nonsmooth operations such as sorting. Consequently, reasoning about rendering requires composing verification techniques across heterogeneous computational layers, a capability absent from existing frameworks.

In this paper, we introduce **Abstract Rendering**, a method for computing the set of all possible images rendered from a continuously varying range of camera poses looking at a scene with a range variations. This set of images, referred to as an **Abstract Image**, is compactly represented using constraints on the image matrix, such as interval or linear bounds. This computed abstract image can then be passed through an existing neural network verification tool [4] to verify models for visual tasks. Our framework is illustrated in Figure 1, with representative results in Figure 4. For example, consider a ResNet classifier downstream of a camera rotating 360° around an airplane in a 3D scene; our method certifies the subrange of viewing angles over which the airplane is classified correctly.

**Our framework**: Abstract Rendering + Downstream Certification

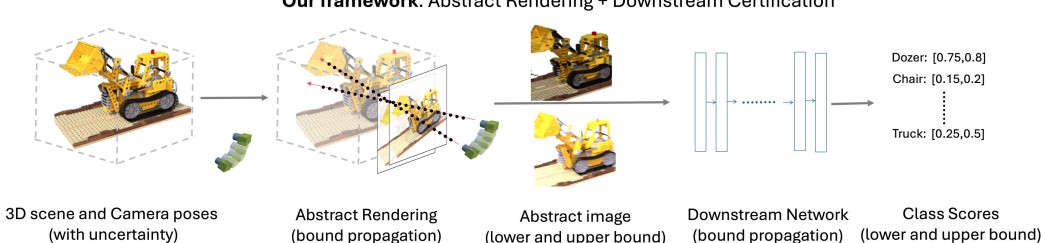

| 3D scene and Camera poses (with uncertainty) | Abstract Rendering (bound propagation) | Abstract image (lower and upper bound) | Downstream Network (bound propagation) | Class Scores (lower and upper bound) |

Figure 1: Pipeline of applying abstract rendering to certify downstream network. A range of camera pose uncertainty in a NeRF scene (Left) is propagated step by step through the rendering process (Left Mid) to produce abstract images (Mid). These abstract images are then passed through the downstream network layer by layer (Right Mid), yielding per-class output bounds (Right). For classification, if the lower bound of the Dozer class exceeds the upper bounds of all other classes, the prediction is certified under the given camera uncertainty.

Our abstract rendering algorithm applies **linear sets** to represent and propagate uncertainties from camera pose and scene parameters, through the rendering pipeline. By decomposing the rendering process into composition of basic operations, we develop linear bound propagation rules that maintain tight output bounds when transforming linear input sets. This compositional approach ultimately produces an abstract image that preserves uncertainty information. To optimize the trade-off between bound precision and computational efficiency, we adopt CROWN [10, 11] for handling standard operations like division and matrix multiplication.

However, certain operators in the rendering process lack native support. To address this, we develop novel linear bound propagation techniques for three rendering-specific operations: matrix inversion, sorting-based aggregation, and cumulative product summation. For matrix inversion required in computing projected 2D Gaussian probability densities, our Taylor series-based approximation produces tighter linear bounds than standard adjugate methods. Sorting-based aggregation, critical for occlusion determination between Gaussians, is reformulated as an equivalent indicator-based process to mitigate over-approximation caused by independently bounding sorted outputs while ignoring index mutual exclusivity. Cumulative product summation, used in opacity/occlusion-weighted value aggregation, employs an ordered strategy that significantly improves bound tightness (particularly beyond 1,000 terms) at moderate computational expense.

In summary, our key contributions are as follows. (1) We introduce the first abstract rendering algorithm to compute abstract images for scenes represented by Gaussian Splats and NeRFs under semantic variations in the 3D world. (2) We develop a unified mathematical and software framework for abstracting rendering pipelines, with novel techniques for matrix inversion, sorting-based aggregation, and cumulative product summation, extending beyond what is supported by existing tools like CROWN. (3) We integrate abstract rendering with neural network verification tool — CROWN, to certify visual tasks such as image classification, pose estimation, and object detection, enabling statements like: "Across camera viewpoints or scene variations, the target model consistently classifies, estimates, or detects object within specified error bounds."

## 2 Related Work

The only closely related work computes abstract images for triangular mesh scenes [6]. To handle camera-pose uncertainty, that method collects each pixel's RGB values into intervals over all possible viewpoints. In practice, this amounts to computing partial interval depth-images for each triangle and merging them using a depth-based union, resulting in an interval image that over-approximates all concrete images for that region of poses. This work considers only camera translations, not rotations. Our method works with Gaussian splatting or NeRF which involve more complex operations and are learnable.

Numerous methods exist in neural network verification [5, 12, 13, 14, 15, 16], including SMT-based approaches [17] and MIP-based techniques [18]. Among these, linear bound propagation has emerged as a prominent strategy, leveraging linear relaxations of nonlinear activations and propagating these bounds layer-wise to efficiently compute output guarantees [19, 11, 20]. Researchers have further advanced the field by introducing complex geometric abstractions such as polyhedra [21], zonotopes [22], and starsets [16, 14]. However, our work diverges conceptually from traditional neural network verification: rather than verifying neural networks themselves, we focus on verifying a broader algorithmic framework that embeds neural networks as components. Crucially, many operations encountered in this context, such as matrix inverse, sorting or cumulative product, are not handled by existing neural network verification tools, necessitating novel theoretical and technical innovations.

Adversarial robustness has been studied most deeply for pixel-space perturbations, where attacks [23, 24, 25, 26, 27] and certificates [11, 20, 14, 16, 28] are constrained to small $l_p$ balls around the original image. Follow-up work broadened the threat model to 2D semantic transforms [29, 30, 31, 32, 33, 34, 35, 36, 37, 38, 39, 40], including global hue or brightness shifts, in-plane rotations, translations, elastic warps, but still treats the scene as a rigid 2D grid. Our method, instead, handles semantic, 3D perturbations that stem from changing the camera pose itself (e.g., orbiting or translating the camera around a fixed object), which jointly affect all pixels in a viewpoint-consistent manner.

Only a handful of papers explore this regime. Athalye [41] synthesizes 3D-printable adversarial objects that fool classifiers over many viewpoints, yet offer no formal guarantee of success across the entire pose distribution. Camera-Motion Smoothing (CMS) by Hu [42] provides the first certificates for small six-DoF camera perturbations via randomized smoothing, but it relies on dense point-cloud supervision and is restricted to narrow pose radii. In Hu's later work [43], he partitions pose space and smooths in the image domain to improve efficiency, yet still yields conservative bounds over only modest displacements. View-Invariant Adversarial Perturbations (VIAP) [44] craft attacks that remain effective across multiple viewpoints, but—as with Athalye et al.—provide no certification. Our approach is the first to deliver tight, end-to-end formal guarantees for standard vision networks across a full 360° camera sweep with realistic rendering.

## 3 Preliminaries: Linear Approximations and Rendering Algorithms

Abstract Rendering bounds uncertainty in rendered images by propagating linear bound through each operation within the rendering pipeline. In this section, we delineate the fundamental operations integral to rendering algorithms and demonstrate their amenability to tight linear over-approximations. Subsequently, we provide an overview of two prominent neural rendering techniques: 3D Gaussian Splatting and Neural Radiance Fields (NeRF).

### 3.1 Linear Over-approximability

**Notations.** For vectors or matrices $x, y$ of the same shape, $x < y$ denotes element-wise comparison. The Frobenius norm of a vector or matrix $x$ is denoted by $\|x\|$. Boldface (e.g., **pc**) indicates set-valued variables, while non-bold (e.g., pc) denotes fixed values. A *linear set* (or polytope) is defined as $\{x \in \mathbb{R}^n \mid Ax \leq b\}$, where $Ax \leq b$ specifies linear constraints. A *piecewise linear relation* $R \subseteq \mathbb{R}^n \times \mathbb{R}^m$ has the form $\left\{(x, y) \mid \underline{A}x + \underline{b} \leq y \leq \overline{A}x + \overline{b}, \ Ax \leq b\right\}$, where $\underline{A}, \overline{A}, \underline{b}, \overline{b}$ define linear bounds on $y$ given $x$. A *constant relation* is a special case where $\underline{A} = \overline{A} = 0$.

We consider a class of functions that can be tightly over-approximated by piecewise linear lower and upper bounds over any compact input domain.

**Definition 1.** *A function $f : S \to \mathbb{R}^m$ is called* linearly over-approximable *if, for any compact $X \subseteq S$ and $\varepsilon > 0$, there exist piecewise linear maps $\ell_X, u_X : X \to \mathbb{R}^m$ such that for all $x \in X$, $\ell_X(x) \le f(x) \le u_X(x)$ and $|u_X(x) - \ell_X(x)| < \varepsilon$.*

**Proposition 1.** *Any continuous function is linearly over-approximable.*

Although Proposition 1 implies piecewise linear bounds on continuous functions can be adjusted sufficiently tight by shrinking the neighborhood, finding accurate bounds over larger neighborhoods remains a significant challenge. Our implementation of abstract rendering employs CROWN [10] for computing such bounds. Except for $\mathrm{Ind}$ and $\mathrm{Sort}$, all the basic operations listed in Table 1 are continuous and therefore linearly over-approximable. Moreover, since $\mathrm{Ind}$ is a piecewise constant function, it can be tightly bounded on each subdomain by partitioning at $x = 0$, allowing it to be treated as a linearly over-approximable function. This is formalized in Corollary 1.

**Corollary 1.** *All operations in Table 1, except for* $\mathrm{Sort}$*, are linearly over-approximable.*

Table 1: Basic Operation Table. Conditional $A?B : C$ returns $B$ if $A$ is true and otherwise $C$. Permuting $x$ based on $y$ means reordering the elements of $x$ according to the indices that sort $y$ in ascending order. E.g., permuting $(9, 3, 7)$ based on $(5, 13, 8)$ results in $(9, 7, 3)$.

| Operation | Inputs | Output | Math Representation |
|---|---|---|---|
| Element-wise add (Add) | $x, y \in \mathbb{R}^{n \times m}$ | $z \in \mathbb{R}^{n \times m}$ | $z_{ij} = x_{ij} + y_{ij}$ |
| Element-wise multiply (Mul) | $x, y \in \mathbb{R}^{n \times m}$ | $z \in \mathbb{R}^{n \times m}$ | $z_{ij} = x_{ij} \cdot y_{ij}$ |
| Division (Div) | $x \in \mathbb{R}_{>0}^{n \times m}$ | $z \in \mathbb{R}_{>0}^{n \times m}$ | $z_{ij} = 1/x_{ij}$ |
| Matrix multiplication (Mmul) | $x \in \mathbb{R}^{n \times m}, y \in \mathbb{R}^{m \times k}$ | $z \in \mathbb{R}^{n \times k}$ | $z = x \times y$ |
| Matrix inverse (Inv) | $x \in \mathbb{R}^{n \times n}, \det(x) \ne 0$ | $z \in \mathbb{R}^{n \times n}$ | $z = x^{-1}$ |
| Matrix power (Pow) | $x \in \mathbb{R}^{n \times n}, k \in \mathbb{N}$ | $z \in \mathbb{R}^{n \times n}$ | $z = x^k$ |
| Summation (Sum) | $x \in \mathbb{R}^n$ | $z \in \mathbb{R}$ | $z = \sum_{i=1}^{k} x_i$ |
| Product (Prod) | $x \in \mathbb{R}^n$ | $z \in \mathbb{R}$ | $z = \prod_{i=1}^{k} x_i$ |
| Matrix transpose ($\top$) | $x \in \mathbb{R}^{n \times m}$ | $z \in \mathbb{R}^{m \times n}$ | $z_{ij} = x_{ji}$ |
| Element-wise exponential (Exp) | $x \in \mathbb{R}^{n \times m}$ | $z \in \mathbb{R}^{n \times m}$ | $z_{ij} = e^{x_{ij}}$ |
| Frobenius norm (Norm) | $x \in \mathbb{R}^{n \times m}$ | $z \in \mathbb{R}_{\ge 0}$ | $z = \|x\|$ |
| Element-wise indicator (Ind) | $x \in \mathbb{R}^{n \times m}$ | $z \in \mathbb{R}^{n \times m}$ | $z_{ij} = (x_{ij} > 0)?1 : 0$ |
| Sorting (Sort) | $z \in \mathbb{R}^n$ | $x \in \mathbb{R}^n, y \in \mathbb{R}^n$ | permute $x$ based on $y$ |

### 3.2 Rendering Algorithms: Gaussian Splat and NeRF

**Scenes and Camera Model** A *3D Gaussian scene* $\mathsf{ScG}$ is defined as a finite collection of 3D Gaussians in world coordinate frame $w$ indexed by $\mathsf{l}$. Each 3D Gaussian is specified by its mean $\mu_\mathsf{w}$, covariance $\Sigma_\mathsf{w}$, opacity $\mathsf{o}$ and RGB color $\mathsf{c}$. For simplicity, we use RGB instead of spherical harmonics to represent 3D Gaussians' colors. A *neural radiance field* $\mathsf{ScN}$ consists of an opacity network $\mathsf{F_o}$ mapping 3D points to opacity, a color network $\mathsf{F_c}$ mapping 3D points and view directions to color, maximum sampling distance $\mathsf{L}$, and sampling count $\mathsf{N}$. A *camera* $C$ is characterized by translation $\mathsf{T} \in \mathbb{R}^3$, rotation $\mathsf{R} \in \mathbb{R}^{3 \times 3}$, focal lengths $(\mathsf{f_x}, \mathsf{f_y}) \in \mathbb{R}^2$, and principal point offsets $(\mathsf{c_x}, \mathsf{c_y}) \in \mathbb{R}^2$.

**Gaussian Splat rendering** Algorithm 1 takes a Gaussian scene $\mathsf{ScG}$, a camera $\mathsf{C}$ and a pixel coordinate $\mathsf{u}$, and outputs the rendered RGB value $\mathsf{pc}$ at pixel $\mathsf{u}$. Rendering proceeds in four steps: (1) it transforms 3D Gaussians from world to camera coordinate $(\mu_\mathsf{c}, \Sigma_\mathsf{c})$ (Line 2-3); (2) it projects 3D Gaussians onto image plane $(\mu_\mathsf{p}, \Sigma_\mathsf{p})$ and computes its weighed probability density $\mathsf{a}$ at pixel $\mathsf{u}$ (Line 4-10); (3) it sorts the Gaussians according to the distance of their means to the image plane (Line 11-14); (4) it computes $\mathsf{pc}$ by aggregating the colors of all Gaussians weighted according to their relative positions (Line 16-18). The full image is rendered by applying GAUSSIANSPLAT at every pixel location.

**NeRF rendering** Algorithm 2 takes as input a neural radiance field $\mathsf{ScN}$, a camera $\mathsf{C}$, and a pixel coordinate $\mathsf{u}$, and outputs the rendered RGB value $\mathsf{pc}$ at that pixel. The rendering proceeds in three steps: (1) it computes the normalized direction of the camera ray $\mathsf{dir_w}$ corresponding to $\mathsf{u}$ and samples $\mathsf{N}$ points $\mathsf{x}$ along the ray within the maximum range $\mathsf{L}$ (Line 1–4); (2) it evaluates the opacity and

**Algorithm 1** GAUSSIANSPLAT(ScG, C, u)

1: **for all** $i \in I$ **do**
2:     $\mu_c[i] \leftarrow R \times (\mu_w[i] - T)$
3:     $\Sigma_c[i] \leftarrow R \times \Sigma_w[i]$
4:     $K[i] \leftarrow \begin{bmatrix} \frac{f_x}{\mu_c[i,2]} & 0 & \frac{c_x}{\mu_c[i,2]} \\ 0 & \frac{f_y}{\mu_c[i,2]} & \frac{c_y}{\mu_c[i,2]} \end{bmatrix}$
5:     $J[i] \leftarrow \begin{bmatrix} \frac{f_x}{\mu_c[i,2]} & 0 & -\frac{f_x \cdot \mu_c[i,0]}{\mu_c^2[i,2]} \\ 0 & \frac{f_y}{\mu_c[i,2]} & -\frac{f_y \cdot \mu_c[i,1]}{\mu_c^2[i,2]} \end{bmatrix}$
6:     $\mu_p[i] \leftarrow K \times \mu_c[i]$
7:     $\Sigma_p[i] \leftarrow J[i] \times \Sigma_c[i] \times J[i]^\top$
8:     $\text{Conic}[i] \leftarrow \text{Inv}(\Sigma_p[i])$
9:     $q[i] \leftarrow (u - \mu_p[i])^\top \times \text{Conic}[i] \times (u - \mu_p[i])$
10:    $a[i] \leftarrow o[i] \cdot \text{Exp}(-\frac{1}{2} \cdot q[i])$
11:    $d[i] \leftarrow \mu_c[i, 2]$
12: **end for**
13: $as \leftarrow \text{Sort}(a, d)$
14: $cs \leftarrow \text{Sort}(c, d)$
15: **for all** $i \in I$ **do**
16:    $oc[i] \leftarrow \prod_{j=1}^{i-1}(1 - as[j])$
17: **end for**
18: $pc \leftarrow \sum_{i=1}^{N}(oc[i] \cdot as[i] \cdot cs[i])$
19: **return** $pc$

**Algorithm 2** NERF(ScN, C, u)

1: $\text{dir}_c \leftarrow \left[ \frac{u_x - c_x}{f_x}, \ \frac{u_y - c_y}{f_y}, \ 1 \right]$
2: $\text{dir}_w \leftarrow \frac{\text{dir}_c \times R^\top}{||\text{dir}_c||}$
3: **for all** $i \in \text{range}(N)$ **do**
4:    $x[i] \leftarrow T + \frac{(2i-1) \cdot L \cdot \text{dir}_w}{2N}$
5:    $c[i] \leftarrow F_c(x[i], \text{dir}_w)$
6:    $o[i] \leftarrow F_o(x[i])$
7:    $a[i] \leftarrow 1 - \text{Exp}\left(-\frac{o[i] \cdot L}{N}\right)$
8:    $oc[i] \leftarrow \prod_{j=1}^{i-1}(1 - a[j])$
9: **end for**
10: $pc \leftarrow \sum_{i=1}^{N}(oc[i] \cdot a[i] \cdot c[i])$
11: **return** $pc$

**Algorithm 3** MATRIXINV(X, $X_{\text{ref}}$, k)

1: $\Delta X \leftarrow -(X - X_{\text{ref}}) \times X_{\text{ref}}^{-1}$
2: **assert** $||\Delta X|| < 1$
3: $Xp \leftarrow \sum_{i=0}^{k} X_{\text{ref}}^{-1} \times \text{Pow}(\Delta X, i)$
4: $X_R \leftarrow ||X_{\text{ref}}^{-1}|| \cdot \frac{||\Delta X||^{k+1}}{1 - ||\Delta X||}$
5: $\text{lXinv} \leftarrow Xp - X_R$
6: $\text{uXinv} \leftarrow Xp + X_R$
7: **return** $\langle \text{lXinv}, \text{uXinv} \rangle$

color of each sampled point using the networks $F_o$ and $F_c$ (Line 5–6); (3) it aggregates the colors along the ray by computing a weighted sum based on the opacity and depth ordering of the samples (Line 7–10).

Having introduced two representative rendering algorithms, we now present a formal definition of abstract rendering.

**Abstract rendering problem.** Our goal is to design an algorithm that for all pixel $u \in WH$, it takes as input a linear set of scenes **Sc** and cameras **C**, and outputs a linear set of pixel color **pc** such that GAUSSIANSPLAT(Sc, C, u) $\in$ **pc** or NERF(Sc, C, u) $\in$ **pc**) for each Sc $\in$ **Sc** and C $\in$ **C**.

## 4 Methodology: Abstract Rendering

Our abstract rendering algorithm processes scene and camera inputs by incrementally building piecewise linear relations between intermediate variables and inputs through standard rendering steps, proceeding until pixel-input relationships are established. These relations enable linear bound computation for each pixel, yielding linear bound for image. Since most GAUSSIANSPLAT and NERF operations are linear, their relations compose directly. In this section, we highlight three rendering-specific nonlinear operations (matrix inverse, sorting-based aggregation and cumulative product summation) and introduce tailored techniques to tightly bound them with linear over-approximations.

### 4.1 Abstracting Matrix Inverse by Taylor Series

Bounding matrix inversion (Line 8 of GAUSSIANSPLAT) via the adjugate method[2] often introduces significant over-approximation due to uncertainty in the denominator. To address this, we propose a Taylor series-based algorithm, MATRIXINV (Algorithm 3), which reformulates matrix inversion using only addition, multiplication, and the inverse of a fixed reference matrix. Although the remainder term still involves division, its influence diminishes as the Taylor order increases.

---

[2]The adjugate method computes the inverse as the adjugate matrix divided by the determinant.

| **Algorithm 4** VR-IND(a, c, d) | **Algorithm 5** SUMCUMPROD(a, c) |
|---|---|
| 1: **for all** i ∈ I **do** | 1: pc ← [0, 0, 0] |
| 2:     $oc[i] \leftarrow \prod_{j=1}^{N} (1 - a[j] \cdot \text{Ind}(d[i] - d[j]))$ | 2: **for all** i ∈ reverse(I) **do** |
| 3: **end for** | 3:     pc ← a[i] · c[i] + (1 − a[i]) · pc |
| 4: $pc \leftarrow \sum_{i=1}^{N} (oc[i] \cdot a[i] \cdot c[i])$ | 4: **end for** |
| 5: **return** pc | 5: **return** pc |

MATRIXINV takes a non-singular matrix X, a non-singular reference matrix $X_{ref}$, and a Taylor order k as input, and returns lower and upper bounds lXinv and uXinv for $X^{-1}$. The algorithm first checks the convergence condition (Line 2), then computes the $k$-th order Taylor approximation and its remainder (Lines 3–4), and finally derives bounds via basic addition and subtraction (Lines 5–6).

**Lemma 1.** *Given any non-singular matrix* X, *the output of Algorithm* MATRIXINV $\langle lXinv, uXinv \rangle$ *satisfies that:* $lXinv \leq \text{Inv}(X) \leq uXinv$.

Lemma 1 guarantees that MATRIXINV yields valid bounds for the Inv operation with fixed input. When extended to a linear set of input matrices, the linear bounds of lXinv and uXinv can be obtained by propagating linear relations line by line. Their union defines the overall linear bounds for the inversion. Example 1 illustrates the advantage of using MATRIXINV over the adjugate method in terms of bound tightness.

**Example 1.** *Given the constant lower and upper bounds of input matrices,*

$$\underline{X} = \begin{bmatrix} 0.60 & -0.2 \\ -0.02 & 0.90 \end{bmatrix}, \quad \overline{X} = \begin{bmatrix} 0.90 & 0.02 \\ 0.02 & 1.30 \end{bmatrix},$$

*we combine lower bound of* lXinv *and upper bound of* uXinv *as overall bounds for* Inv *operation, and take* $\|uXinv - lXinv\|$ *as measurement for bound tightness. The adjugate method yields a bound width of* 1.22, *while* MATRIXINV *reduces it to* 0.70, *closely matching the empirical value of* 0.66.

### 4.2 Abstracting Sorting-based Summation by Indicator

The sorting step in Lines 13–14 of GAUSSIANSPLAT models occlusion: a front Gaussian occludes those behind, but not vice versa. Bounding sorted elements independently introduces large over-approximations in the final pixel color (Line 18), as it neglects the inherent ordering constraint. To mitigate this, we introduce VR-IND (Algorithm 4), which replaces sorting with pairwise indicator-based occlusion modeling. This approach preserves the output while significantly improving bound tightness in linear relaxation.

**Lemma 2.** *For any given inputs — effective opacities* a, *colors* c *and depth* d, VR-IND *outputs the same value as computation from Line 13 to Line 18 in* GAUSSIANSPLAT.

Lemma 2 confirms that VR-IND produces the same output as Lines 13–18 in GAUSSIANSPLAT, capturing both occlusion (Line 2) and color aggregation (Line 4) without explicit sorting. Example 2 further demonstrates its advantage in reducing over-approximation.

**Example 2.** *Consider a scene with three Gaussians (red, green, blue) and a camera at the origin facing upward, with each coordinate perturbed by ±0.3. As shown in Figure 2, the upper bound obtained via linear relaxation through* VR-IND *is significantly tighter than that derived from direct bound propagation through* GAUSSIANSPLAT. *An analytical example is given in Appendix A.*

### 4.3 Abstract Summation of Cumulative Product

The summation of cumulative products arises in Lines 8–10 of NERF, where occlusion effects are computed and colors are aggregated to produce the final pixel color. Naively bounding each element of oc independently and then aggregating can lead to significant over-approximation, since the intermediate terms a are shared across multiple entries. To mitigate this, we propose a factorization-based formulation, SUMCUMPROD (Algorithm 5), which computes each element of a only once in the final expression for pc. This approach preserves exactness for fixed inputs while enabling substantially tighter bounds during linear relaxation.

**Lemma 3.** *For any given inputs — effective opacities* a, *colors* c, SUMCUMPROD *outputs the same value as computation from Line 8 to Line 10 in* NERF.

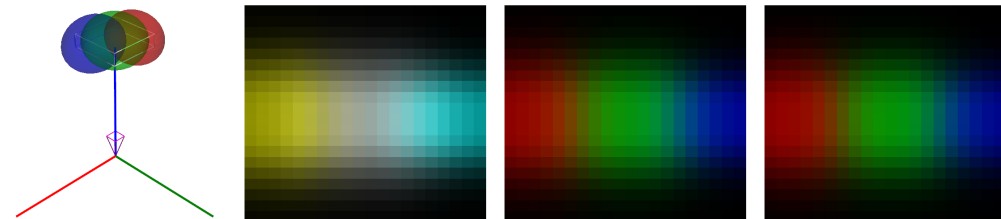

Figure 2: Example 2 (left) shows the three-Gaussian scene. The corresponding upper bounds are visualized as follows: via GAUSSIANSPLAT (mid-left), via VR-IND (mid-right), and via sampling (right). Notably, the bounds produced by VR-IND are tighter (darker) and closely align with the sampled results.

Lemma 3 confirms that, for fixed input, SUMCUMPROD exactly reproduces the original NERF output without explicitly computing occlusion for each sampled point. Example 3 shows that it produces much tighter bounds than the standard method. However, SUMCUMPROD has two limitations: (1) its recursive structure necessitates iterative bound propagation, thereby increasing computational cost; (2) it assumes a fixed order of a, restricting its applicability to Gaussian-represented scenes where the Gaussian depth order remains unchanged, such as in scenarios involving camera translation uncertainty.

**Example 3.** *Consider a single-channel color* $c = [0.8, 1.0, 0.9, 0.8, 0.5]$ *and effective opacity* $a = [0.2, 0.5, 0.6, 0.8, 0.1]$ *with* $\pm 0.1$ *perturbations on each* $a[i]$. SUMCUMPROD *yields a tight bound on output pixel color,* $[0.826, 0.926]$, *versus a looser* $[0.780, 1.029]$ *from Lines 8–10 of* NERF. *Sampling 10,000 random perturbations gives* $[0.831, 0.922]$, *closely matching the result from* SUMCUMPROD.

### 4.4 Linear Approximability of ABSTRACTRENDER

The operations in MATRIXINV, VR-IND, and SUMCUMPROD fall within those listed in Table 1. Since sorting can be eliminated using VR-IND, and all remaining operations are linear over-approximable, we arrive at Theorem 1. It states that both GAUSSIANSPLAT and NERF can be rewritten using only linearly approximable operations, enabling computation of tight bounds via input domain partitioning.

**Theorem 1.** *By replacing components of* GAUSSIANSPLAT *and* NERF *with* MATRIXINV, VR-IND, *and* SUMCUMPROD, *both rendering algorithms are fully linear over-approximable.*

## 5 Experiments with Abstract Rendering

We implemented the abstract rendering algorithm ABSTRACTRENDER described in Section 4 with both GAUSSIANSPLAT and NERF. As mentioned earlier, the linear approximation of the continuous operations in Table 1 is implemented using CROWN [10].

**Scene Description.** We evaluate ABSTRACTRENDER on scenes of varying scales and complexities, represented using GAUSSIANSPLAT and NERF. The scenes include: `Lego`, `Chair`, and `Drums` [2], which are single-object scenes on empty backgrounds; `PineTree` [6], a synthetic boulevard scene with trees; `Airport` [8], a large-scale photorealistic airport environment; `Garden` [1], a real-world scene; and `Airplane`, `Truck`, and `Car`, containing objects corresponding to CIFAR-10 classes. GAUSSIANSPLAT reconstructions are generated using Splatfacto [45], while those of NERF are trained with the vanilla NERF algorithm [2], both implemented with standard Nerfstudio settings. Further details of these scenes are provided in the appendix.

**Experimental Setup.** For each scene, we evaluate ABSTRACTRENDER under varying camera poses and input perturbations. As a baseline, we construct **empirical bound images** by (1) sampling 50 images per input partition (the full perturbation range is divided into hundreds or thousands of partitions) and (2) computing pixel-wise lower and upper bounds across the samples. Since exhaustive enumeration is infeasible, these empirical bound images are **under-approximations** of the exact set of renderable images, whereas our framework provides sound **over-approximations**. The proximity of ABSTRACTRENDER results to the empirical bound images indicates the tightness of our computed bounds.

**Metrics and Evaluation.** To measure the tightness between the pixel-wise lower and upper bound images, we report **Mean Pixel Gap** (MPG) and **Max Pixel Gap** (XPG), defined as:

$$\text{MPG} = \frac{1}{|\text{WH}|} \sum_{j \in \text{WH}} \|\overline{\text{pc}_j} - \underline{\text{pc}_j}\|, \quad \text{XPG} = \max_{j \in \text{WH}} \|\overline{\text{pc}_j} - \underline{\text{pc}_j}\|$$

where $\overline{\text{pc}_j}$ and $\underline{\text{pc}_j}$ denote the pixel-wise upper and lower bounds, and WH represents the set of all pixel coordinates on the image plane. Smaller values of MPG or XPG indicate a smaller difference between the lower and upper bound images. Table 2 summarizes the results across different scene representations and perturbations, while Figure 3 visualizes the comparison between ABSTRACTRENDER results and empirical bounds for two scenes: `Airplane` and `Lego`.

Table 2: ABSTRACTRENDER results for scenes represented by GAUSSIANSPLAT (GS) and NERF, along with empirical bounds. CPR: Camera Perturbation Range; Dim: Perturbation Dimension; Res: Rendered Image Resolution; Rt: Runtime (min); MPG: mean pixel gap; XPG: maximum pixel gap.

| Scene | CPR | Dim | Res | GS | | | NeRF | | | Empirical | |
|---|---|---|---|---|---|---|---|---|---|---|---|
| | | | | Rt | MPG | XPG | Rt | MPG | XPG | MPG | XPG |
| Lego | 0.1 (rad) | yaw | 80×80 | 22 | 0.51 | 1.73 | 25 | 0.40 | 1.73 | 0.22 | 1.57 |
| Chair | 0.1 (rad) | yaw | 50×50 | 17 | 0.46 | 1.73 | 20 | 0.40 | 1.73 | 0.40 | 1.72 |
| Drums | 0.1 (m) | x | 50×50 | 5.3 | 0.19 | 1.62 | 5.8 | 0.16 | 1.55 | 0.13 | 1.54 |
| PineTree | 2 (m) | x | 72×72 | 4.1 | 0.27 | 1.73 | 4.9 | 0.23 | 1.73 | 0.06 | 1.37 |
| PineTree | 10 (m) | x | 72×72 | 24 | 0.47 | 1.73 | 28 | 0.36 | 1.73 | 0.18 | 1.37 |
| Airport | 0.027 (rad) | roll | 160×160 | 27 | 0.56 | 1.69 | 35 | 0.42 | 1.63 | 0.21 | 1.38 |
| Airport | 0.03 (rad) | roll | 160×160 | 30 | 0.59 | 1.70 | 43 | 0.51 | 1.68 | 0.22 | 1.41 |
| Garden | 0.5 (m) | x | 100×100 | 20 | 0.53 | 1.59 | 11 | 0.37 | 1.53 | 0.34 | 1.21 |
| Airplane | 2 (m) | x | 80×80 | 16 | 0.41 | 1.68 | 19 | 0.33 | 1.61 | 0.15 | 1.39 |

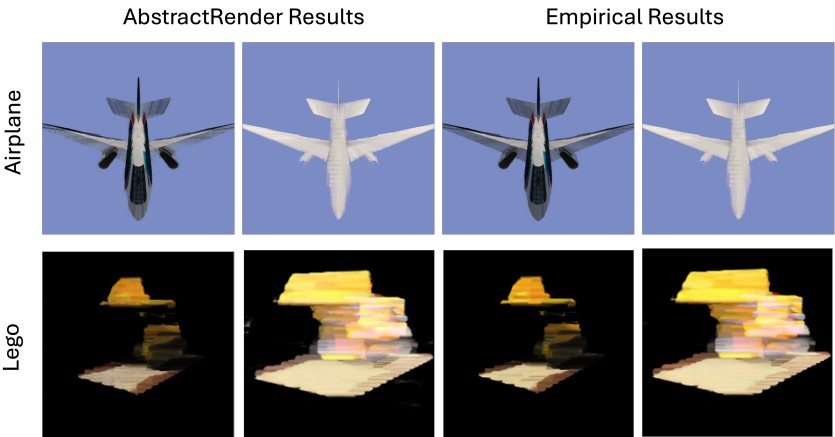

Figure 3: Comparison of lower and upper images produced by ABSTRACTRENDER (Left) and empirical approach (Right). The `Airplane` scene (Top) is represented by GAUSSIANSPLAT, while the `Lego` scene (Bottom) is represented by NERF. The experimental settings for both cases follow those reported in Table 2.

From Table 2, we see that ABSTRACTRENDER is applicable to diverse scenes, including single-object (`Lego`), synthetic (`PineTree`), and large-scale realistic (`Airport`, `Garden`) scenes, with reasonable runtime at $200 \times 200$ resolution. The framework supports camera perturbations in both position (x, y, z) and orientation (roll, pitch, yaw). The metrics (MPG and XPG) computed by ABSTRACTRENDER (under GS and NeRF columns in Table 2) are larger than the corresponding empirical bounds, as expected, since our framework produces over-approximations, which inherently encompass a larger image space than the under-approximated empirical bounds. We further observe that bounds computed for NERF exhibit tighter margins than those for GAUSSIANSPLAT, as indicated by smaller MPG/XPG gaps relative to empirical samples. This is because NERF can leverage our linear bound

relaxation for cumulative product summation (Section 4.3), whereas GAUSSIANSPLAT cannot, due to variations in the depth of 3D Gaussians on the image plane under camera rotation uncertainty.

Overall, these experimental results confirm that our method produces sound over-approximations that are meaningfully tight for rendered images from both GAUSSIANSPLAT and NERF under camera pose uncertainty.

**ABSTRACTRENDER for Certified Classification** This experiment evaluates ABSTRACTRENDER for certifying neural classifier robustness against camera pose perturbations. We test whether a pretrained CIFAR-10 ResNet [46] maintains correct predictions as the camera orbits the target object azimuthally over 360° at fixed distance and elevation. The 360° range is partitioned into angular intervals, and ABSTRACTRENDER computes abstract images for each, which are propagated through the classifier via CROWN [10] for set-based certification. An interval is robust if the lower bound of the target label exceeds the upper bounds of all other classes. Verification coverage is the percentage of certifiably robust intervals. Table 3 shows results for Airplane, Car, and Truck in GAUSSIANSPLAT and NERF scenes. Figure 4 visualizes camera regions where the classifier is certifiably correct (green) and regions without such guarantees (red). Regions capturing more of the object or its distinctive features are more likely to be verified as robust, e.g., the lateral views of the airplane and car, and the front-left view of the truck.

Table 3: Classifier certification results. Obj: Target Object; SR: Scene Representation, either GAUSSIANSPLAT or NERF; d: Distance to Object Centroid (m); h: Camera Height to Object's Horizon Plane (m); Npart: Number of Partitions; PCP-S: Percentage of Correct Partitions via Sampling; PCP-V: Percentage of Certified Partitions via Formal Verification; Rt: Runtime (min).

| Obj | SR | d | h | Npart | PCP-S | PCP-V | Rt |
|---|---|---|---|---|---|---|---|
| Airplane | GS | 40 | 10 | 62832 | 96.0% | 74.5% | 1140.5 |
| Car | GS | 8 | 2.5 | 62832 | 96.3% | 76.7% | 879.6 |
| Truck | GS | 4 | 1.2 | 62832 | 71.8% | 31.9% | 963.4 |
| Airplane | NERF | 40 | 36 | 3142 | 25.6% | 22.8% | 94.3 |
| Car | NERF | 4 | 0.5 | 3142 | 75.9% | 42.1% | 118.6 |

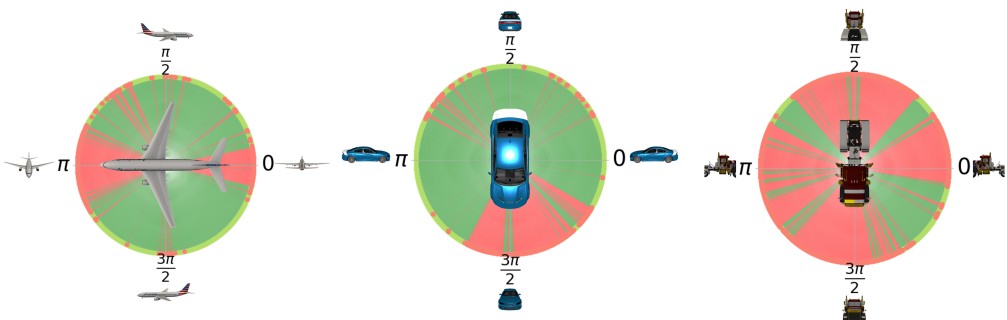

Figure 4: Classifier certification results for Airplane, Car, and Truck in GAUSSIANSPLAT scenes across 0–360° camera rotations. Green: certified regions—images captured within these camera positions are always predicted correctly. Red: uncertified regions—images captured within these camera positions may be predicted incorrectly.

**ABSTRACTRENDER for certified pose estimation** This experiment evaluates ABSTRACTRENDER for certifying the robustness of neural pose estimators against camera pose perturbations. We assess a target pose estimator built upon GateNet [47] (fine-tuned on our airplane and truck datasets) by partitioning the camera's translational perturbation range into intervals. ABSTRACTRENDER generates abstract images capturing all photometric variations within each interval, which are propagated through the estimator via CROWN to verify whether positional errors between pose estimation and ground truth holds below a given error tolerance (20% divation from ground truth). Verification coverage is measured as the proportion of robust intervals across the workspace. Table 4 reports results

for `Airplane` and `Truck` in GAUSSIANSPLAT and NERF scenes, while Figure 5 visualizes camera regions where the estimator meets the error tolerance (green) versus regions without guarantees (red). Regions where the target object occupies a substantial portion of the rendered image are easier to certify—for example, distant regions in GAUSSIANSPLAT scenes and closer regions in NERF scenes, reflecting the different default camera-object distances in the two settings.

Table 4: Pose estimator and object detecter certification results. Obj: Target Object; SR: Scene Representation, either GAUSSIANSPLAT or NERF; d: Distance to Object Centroid; CPR: Camera Perturbation Range (m); PCP-S: Percentage of Correct Partitions via Sampling; PCP-V: Percentage of Certified Partitions via Formal Verification; Thr: Threshold; Rt: Runtime(min).

| Obj | SR | Certified Pose Estimation | | | | | Certified Object Detection | | | | |
|-----|----|----|-----|-------|-------|-------|-----|-----|-------|-------|-------|
| | | d | CPR | PCP-S | PCP-V | Rt | Thr | CPR | PCP-S | PCP-V | Rt |
| Airplane | GS | 174.4 | 6 | 52.4% | 23.3% | 362.2 | 0.45 | 6 | 81.5% | 51.4% | 422.2 |
| Truck | GS | 7.8 | 1 | 37.6% | 22.2% | 311.3 | 0.46 | 1 | 47.2% | 42.4% | 367.3 |
| Airplane | NERF | 40 | 20 | 46.7% | 37.2% | 82.4 | 0.12 | 3.5 | 92.2% | 80.4% | 121.3 |
| Truck | NERF | 8 | 2 | 75.1% | 70.0% | 53.4 | 0.12 | 1.5 | 100.0% | 71.0% | 66.5 |

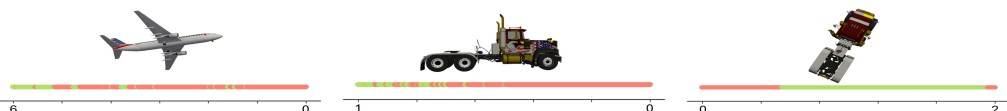

Figure 5: Pose estimator certification results for `Airplane` and `Truck` in GAUSSIANSPLAT scenes (Left, Middle) and `Truck` in a NERF scene (Right) across a linear camera translation. Green: certified regions—camera positions where estimated poses always remain within given error tolerance. Red: uncertified regions—camera positions where estimated poses may exceed given error tolerance.

**ABSTRACTRENDER for certified object detection** This experiment evaluates ABSTRACTREN-DER for certifying neural object detectors under camera pose perturbations. We test a pretrained detector based on the YOLO2 architecture [48], with a backbone from the VNN-COMP 2023 benchmark[3]. The goal is to assess whether the detector can consistently produce bounding boxes that exceeds a predefined confidence threshold under camera translations. The camera's translational range is partitioned into small intervals. For each interval, ABSTRACTRENDER computes abstract images. These images are propagated through the YOLO2 detector. An interval is considered robust if the detector maintains a high-confidence bounding box across all images rendered from that interval. Table 4 reports results for `Airplane` and `Truck` in GAUSSIANSPLAT and NERF scenes, demonstrating ABSTRACTRENDER's capacity to certify object detection models.

## 6 Conclusions

**Limitations.** ABSTRACTRENDER currently has several limitations: First, like other verification approaches, it takes more time than sampling-based approaches. Second, the correctness of the analysis assumes the reconstructed scene accurately represents reality. Third, our framework computes bounds for each pixel's color independently, ignoring correlations between neighboring pixels.

Despite these limitations, ABSTRACTRENDER is the first framework for computing abstract images of scenes represented by Gaussian Splats and NeRF under camera pose or scene uncertainty. It is built using novel linear relational approximations of three rendering-specific operations. By integrating ABSTRACTRENDER with CROWN, we have enabled certification of visual tasks with respect to semantic variations in 3D environments. A promising direction for future work is to accommodate scenes inaccuracies and to certify visual control functions.

---

[3]https://github.com/VNN-COMP/vnncomp2023_benchmarks

## Acknowledgments and Disclosure of Funding

Chenxi Ji, Yangge Li, and Sayan Mitra are supported by a research grant from The Boeing Company and NSF (FMITF-2525287). Huan Zhang and Xiangru Zhong are supported in part by the AI2050 program at Schmidt Sciences (AI2050 Early Career Fellowship) and NSF (IIS SLES-2331967, CCF FMITF-2525287).

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

## A    An analytical example illustrating the advantage of Algorithm VR-IND

Consider two purely white Gaussians (with color values set to 1) whose probability densities and depth to the target pixel are given by $a = \{0.8, 0.5\}$ and $d = \{x, x + 1\}$, respectively, where $x \in [-1, 1]$. We aim to compute the worst-case bound of volumetric blending as follows:

$$as = sort(a, d)$$
$$oc[0] = 1$$
$$oc[1] = 1 - as[0]$$
$$pc = oc[0] \times as[0] + oc[1] \times as[1]$$

we first determine the output range of $pc$ via Naive interval approach. By plugging in range of $x$ into the expression of $d$, we have $d[0] = x \in [-1, 1]$, and $d[1] = x + 1 \in [0, 2]$. Since the relative ordering between $d[0]$ and $d[1]$ cannot be inferred from their interval bound, we must consider both possible cases: one where $(as[0], as[1]) = (a[0], a[1])$, and the other where the assignment is reversed. Thus, we have

$$as[0], as[1] \in [0.5, 0.8];$$
$$oc[1] \in [0.2, 0.5]$$

Hence, the resulting interval bound for $pc$ is:

$$pc \in [1 \cdot 0.5 + 0.2 \cdot 0.5, 1 \cdot 0.8 + 0.5 \cdot 0.8] = [0.6, 1.2]$$

Next, we determine the output range of $pc$ via proposed algorithm VR-IND, in which the same computation is encoded symbolically to account for value-dependent assignments. The volumetric blending procedure is reformulated using indicator functions (Ind), making the dependencies on $d$ explicit:

$$oc[0] = (1 - a[0] \times Ind(d[0] - d[0])) \times (1 - a[1] \times Ind(d[0] - d[1]))$$
$$oc[1] = (1 - a[0] \times Ind(d[1] - d[0])) \times (1 - a[1] \times Ind(d[1] - d[1]))$$
$$pc = oc[0] \times a[0] + oc[1] \times a[1]$$

Given $d[0]$ and $d[1]$, we can compute the values of indicator functions Ind in the above formulation.

$$Ind(d[0] - d[1]) = Ind(x - (x + 1)) = Ind(-1) = 0$$
$$Ind(d[1] - d[0]) = Ind((x + 1) - x) = Ind(1) = 1$$
$$Ind(d[1] - d[1]) = Ind(d[0] - d[0]) = Ind(0) = 0$$

Substituting these indicator values yields:

$$oc[0] = (1 - 0.8 \times 0) \times (1 - 0.5 \times 0) = 1$$
$$oc[1] = (1 - 0.8 \times 1) \times (1 - 0.5 \times 0) = 0.2$$
$$pc = 1 \times 0.8 + 0.2 \times 0.5 = 0.9$$

Hence, using VR-IND, we obtain a precise bound $pc \in [0.9, 0.9]$, which is significantly tighter than the interval-based bound of $[0.6, 1.2]$. Remark: in the real verification pipeline, VR-INDproduces a linear relaxation set instead of a fixed value. Here we use a simplified case for highlighting VR-IND's advantage in terms of bound tightness.

The naive interval method leads to loose bounds due to the loss of dependency information caused by Sort operation. In contrast, the proposed VR-Ind method preserves input-dependent relationships and yields significantly tighter bounds.

# B Proofs

**Proposition 1.** *Any continuous function is linearly over-approximable.*

*Proof.* We present a simple proof for scalar $f$ and this can be extended to higher dimensions. We fix $x_0 \in \mathbb{R}$ and a finite slope $k \in \mathbb{R}$. Since $f(x) - k(x - x_0)$ is continuous, for any $\varepsilon > 0$, there must exist an open neighborhood $U(x_0)$ around $x_0$, such that $\forall x \in R$,

$$|f(x) - f(x_0) - k(x - x_0)| < \frac{\varepsilon}{3}. \tag{1}$$

Equivalently,

$$f(x_0) + k(x - x_0) - \frac{\varepsilon}{3} < f(x) < f(x_0) + k(x - x_0) + \frac{\varepsilon}{3}. \tag{2}$$

We construct candidate linear upper and lower bounds over $U$ as:

$$l_U(x) = k(x - x_0) + f(x_0) - \frac{\varepsilon}{3}, \quad u_U(x) = k(x - x_0) + f(x_0) + \frac{\varepsilon}{3}. \tag{3}$$

We can check that these linear functions $l_U(x)$ and $u_U(x)$ are valid lower and upper bounds for $f(x)$ over $U$, because $\forall x \in U(x_0)$:

$$f(x) - l_U(x) = f(x) - (k(x - x_0) + f(x_0) - \frac{\varepsilon}{3}) > -\frac{\varepsilon}{3} + \frac{\varepsilon}{3} = 0, \tag{4}$$

$$f(x) - u_U(x) = f(x) - (k(x - x_0) + f(x_0) + \frac{\varepsilon}{3}) < \frac{\varepsilon}{3} - \frac{\varepsilon}{3} = 0. \tag{5}$$

Additionally, difference between $l_U(x)$ and $u_U(x)$ is tightly bounded, as $\forall x \in U(x_0)$:

$$|u_U(x) - l_U(x)| = (k(x - x_0) + f(x_0) + \frac{\varepsilon}{3}) - (k(x - x_0) + f(x_0) - \frac{\varepsilon}{3}) = \frac{2\varepsilon}{3} < \varepsilon.$$

Now, for any compact set $B$, the collection $\mathcal{U} = \{U(x) \mid x \in B\}$ forms an open cover of $B$. By the definition of compactness, there exists a finite subcover of $B$, denoted by $\{U_i = U(x_i) \mid i = 1, \ldots, s\}$. We can now define piecewise linear functions on $B$ as follows:

$$\begin{aligned} u_B(x) &= \min_{j \in I} u_{U_j}(x), \\ l_B(x) &= \max_{j \in I} l_{U_j}(x), \quad if\ x \in \{U_j \mid j \in I\} \end{aligned} \tag{6}$$

where $I$ is the index set of $U_i$ that cover $x$. Consequently, we have

$$u_B(x) = \min_{j \in I} u_{U_j}(x) \geq f(x) \geq \max_{j \in I} l_{U_j}(x) = l_B(x), \tag{7}$$

and

$$|u_B(x) - l_B(x)| \leq u_{U_j}(x) - l_{U_j}(x) \leq \varepsilon, \quad \forall j \in I. \tag{8}$$

Thus, $f$ is linearly over-approximable. $\qquad \square$

**Lemma 1.** *Given any non-singular matrix* $\mathsf{X}$*, the output of Algorithm* MATRIXINV $\langle \mathsf{lXinv}, \mathsf{uXinv} \rangle$ *satisfies that:*

$$\mathsf{lXinv} \leq \mathrm{Inv}(\mathsf{X}) \leq \mathsf{uXinv}$$

*Proof.* For any non-singular $n \times n$ matrices $\mathsf{X}$ and $\mathsf{X_{ref}}$, denote their difference as $\Delta \mathsf{X} = -(\mathsf{X} - \mathsf{X_{ref}})$. The $\mathsf{k^{th}}$ order Taylor Polynomial $\mathsf{Xp}$ and remainder $\mathsf{X_R}$ of matrix inverse of $\mathsf{X}$, estimated at $\mathsf{X_{ref}}$, can be written as follows:

$$\mathsf{Xp} = \mathsf{X_{ref}^{-1}} \sum_{i=0}^{k} (\Delta \mathsf{X} \cdot \mathsf{X_{ref}^{-1}})^i \tag{9}$$

$$\mathsf{X_R} = \mathsf{X_{ref}^{-1}} \sum_{i=k+1}^{\infty} (\Delta \mathsf{X} \cdot \mathsf{X_{ref}^{-1}})^i \tag{10}$$

Assuming that $\|\Delta X \cdot X_{ref}^{-1}\| < 1$, the remainder $X_R$ can be bounded by:

$$\|X_R\| \leq \|X_{ref}^{-1}\| \cdot \sum_{i=k+1}^{\infty} \|\Delta X \cdot X_{ref}^{-1}\|^i = \|X_{ref}^{-1}\| \cdot \frac{\|\Delta X \cdot X_{ref}^{-1}\|^{k+1}}{1 - \|\Delta X \cdot X_{ref}^{-1}\|} \tag{11}$$

Using the bound in inequality 11, we obtain bounds on $X^{-1}$:

$$Xp - X_R \leq X^{-1} \leq Xp + X_R \tag{12}$$

Given the definition of $lXinv = Xp - X_R$ and $uXinv = Xp + X_R$, we obtain the lower and upper bound for $X^{-1}$ as:

$$lXinv \leq X^{-1} \leq uXinv. \tag{13}$$

$\square$

**Lemma 2.** *For any given inputs — effective opacities* a, *colors* c *and depth* d, VR-IND *outputs the same value as computation from Line 13 to Line 18 in* GAUSSIANSPLAT.

*Proof.* We first derive the math expression of pixel color pc from GAUSSIANSPLAT. By denoting $\arg\mathrm{sort}(d)$ as a mapping $\sigma : I \to I$, where $I$ refers to the index set of Gaussians, as at Line 13, cs at Line 14 and oc at Line 16 can be written as follows.

$$as_i = a_{\sigma(i)} \tag{14}$$

$$cs_i = c_{\sigma(i)} \tag{15}$$

$$oc_i = \prod_{j=1}^{i-1} 1 - a_{\sigma(j)} \tag{16}$$

According to the definition of Ind operation, we have:

$$\mathrm{Ind}(d_i - d_j) = \begin{cases} 1 & \text{if } d_i > d_j \\ 0 & \text{if } d_i \leq d_j. \end{cases} \tag{17}$$

Then by replacing $i, j$ with $\sigma(i), \sigma(j)$, and multiplying $a_{\sigma(i)}$, (17) becomes:

$$a_{\sigma(i)} \cdot \mathrm{Ind}(d_{\sigma(i)} - d_{\sigma(j)}) = \begin{cases} a_{\sigma(i)} & \text{if } d_{\sigma(i)} > d_{\sigma(j)} \\ 0 & \text{if } d_{\sigma(i)} \leq d_{\sigma(j)} \end{cases} \tag{18}$$

For any fixed index i, by multiplying all the case in the second branch of (18), it follows:

$$\prod_{j=i}^{N}(1 - a_{\sigma(i)} \cdot \mathrm{Ind}(d_{\sigma(i)} - d_{\sigma(j)})) = 1. \tag{19}$$

By applying Equation 16, Equation 18 and Equation 19 into Line 16, $oc_i$ can be written as follows:

$$\begin{aligned} oc_i &= \prod_{j=1}^{i-1}(1 - a_{\sigma(i)}) \\ &= \prod_{j=1}^{i-1}(1 - a_{\sigma(i)} \cdot \mathrm{Ind}(d_{\sigma(i)} - d_{\sigma(j)})) \\ &= \prod_{j=1}^{N}(1 - a_{\sigma(i)} \cdot \mathrm{Ind}(d_{\sigma(i)} - d_{\sigma(j)})) \end{aligned} \tag{20}$$

Then by applying Equation 20 into Line 18, the pixel color pc computed via Algorithm BLENDSORT can be expressed as:

$$pc = \sum_{i=1}^{N}\left(\prod_{j=1}^{N}(1 - a_{\sigma(i)} \cdot \mathrm{Ind}(d_{\sigma(i)} - d_{\sigma(j)}))a_{\sigma(i)}c_{\sigma(i)}\right) \tag{21}$$

Since the summation and product operations are invariant to the reordering of input elements, and both indices $i$ and $j$ iterate over the entire index set $I$, the reordering mapping $\sigma$ in Equation 21 can be eliminated, resulting in:

$$pc = \sum_{i=1}^{N} \left( \prod_{j=1}^{N} (1 - a_i \cdot \mathrm{Ind}(d_i - d_j)) a_i c_i \right) \tag{22}$$

The last equation comes from 17.

Next, we derive the math expression for $pc$ in Algorithm VR-IND. $oc_i$ at Line 2 can be written as:

$$oc_i = \prod_{j=1}^{N} (1 - a_i \cdot \mathrm{Ind}(d_i - d_j)) \tag{23}$$

Then, by applying Equation 23 into Line 4 of Algorithm VR-IND, the expression for $pc$ becomes:

$$pc = \sum_{i=1}^{N} \left( \prod_{j=1}^{N} (1 - a_i \cdot \mathrm{Ind}(d_i - d_j)) a_i c_i \right) \tag{24}$$

Note that the expressions for $pc$ are identical in both Equation 22 and Equation 24. This demonstrates that Algorithms BLENDSORT and VR-IND yield the same input-output relationship while applying different operations, thus completing this proof. $\square$

# C Supplementary Experiment Results

## C.1 Detailed Scenario Information

Table 5 summarizes scene configurations and rendering quality metrics for the scenarios discussed in Section 5. Figure 6 depicts representative GAUSSIANSPLAT-rendered images for each scene.

Table 5: Reconstruction quality for scenes are evaluated by the following metrics: NGauss: Number of Gaussians used for scene representation, applicable only to Gaussian Splatting; PSNR: Peak Signal-to-Noise Ratio; SSIM: Structural Similarity Index Measure; LPIPS: Learned Perceptual Image Patch Similarity.

| Scene | Gaussian Splatting | | | | NeRF | | |
|---|---|---|---|---|---|---|---|
| | NGauss | PSNR | SSIM | LPIPS | PSNR | SSIM | LPIPS |
| Lego | 43543 | 25.60 | 0.95 | 0.07 | 21.38 | 0.81 | 0.19 |
| Chair | 46108 | 22.98 | 0.94 | 0.08 | 22.56 | 0.89 | 0.11 |
| Drums | 50877 | 21.19 | 0.89 | 0.10 | 19.46 | 0.81 | 0.23 |
| Pinetree | 113368 | 31.06 | 0.97 | 0.06 | 22.41 | 0.79 | 0.20 |
| Airport | 617371 | 18.83 | 0.84 | 0.33 | 20.83 | 0.72 | 0.22 |
| Garden | 524407 | 18.74 | 0.37 | 0.32 | 22.15 | 0.80 | 0.20 |
| Plane | 51316 | 28.05 | 0.95 | 0.11 | 22.89 | 0.70 | 0.26 |
| Truck | 47895 | 24.70 | 0.94 | 0.09 | 23.53 | 0.75 | 0.19 |
| Car | 34699 | 26.98 | 0.93 | 0.10 | 20.75 | 0.69 | 0.27 |

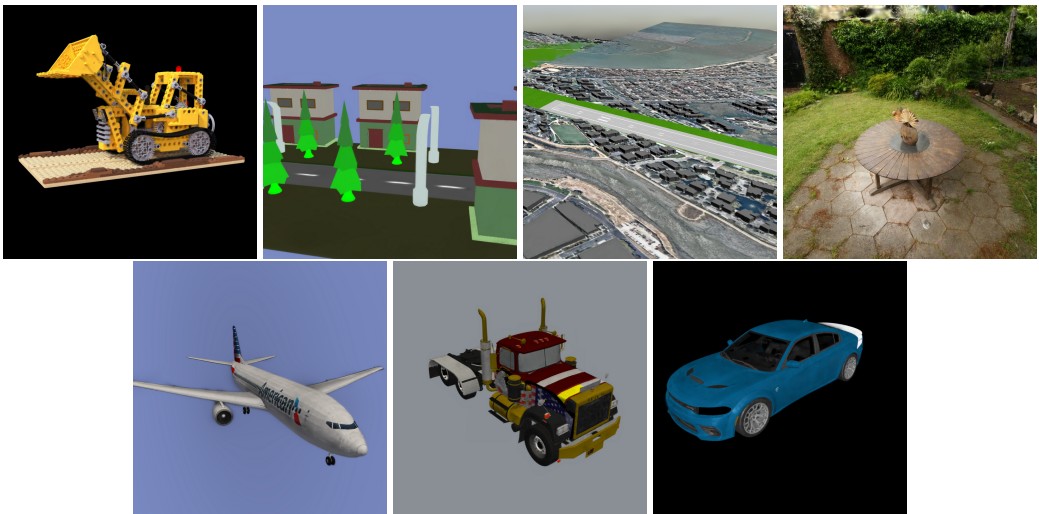

Figure 6: Example rendered image for scenario (left to right, top to bottom): Lego, Pinetree, Airport, Garden, Plane, Truck, Car by GAUSSIANSPLAT.

## C.2 Supplementary Results for Certified Classification

Figure 7 compares certified classification regions with those verified by sampling.

## C.3 Supplementary Results for Certified Pose Estimation

Figure 8 compares certified estimated pose regions with those verified by sampling.

## C.4 Supplementary Results for Certified Object Detection

Figure 9 compares certified detection with those verified by sampling.

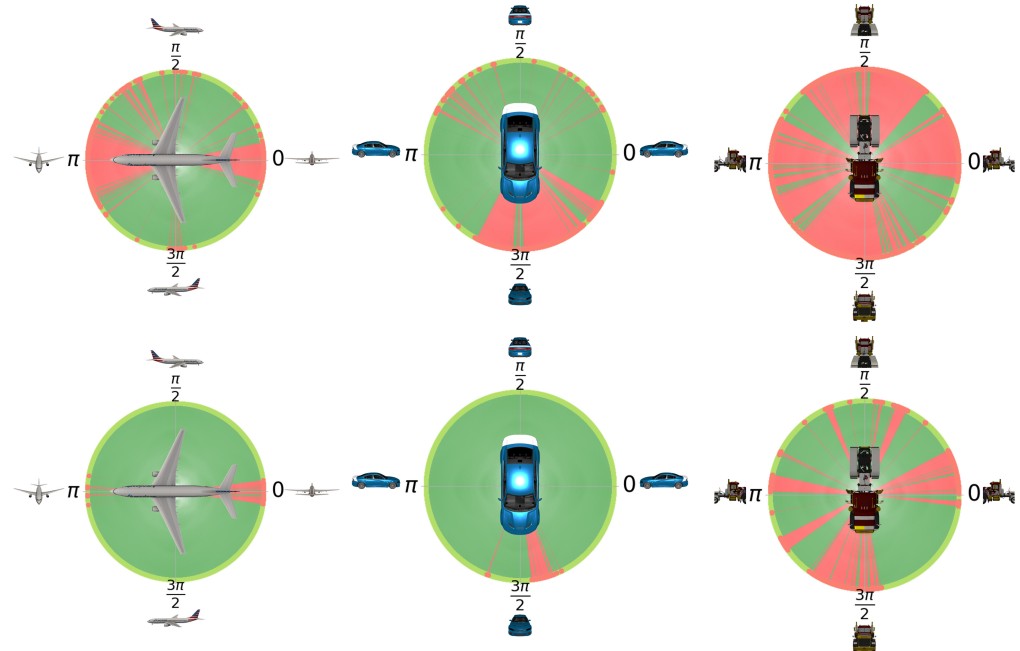

Figure 7: Classifier certification result for AIRPLANE, CAR, and TRUCK by GAUSSIANSPLAT across 0-360° camera rotation. Top row: Certified classification result. Bottom row: Classification result obtained via sampling. Green: Certified/Correct Camera Pose Region. Red: Uncertified/Incorrect Camera Pose Region.

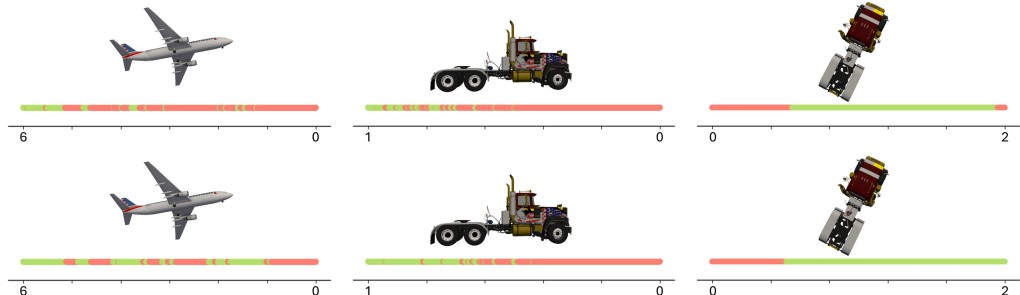

Figure 8: Pose estimation certification results for `Airplane`, `Truck` by GAUSSIANSPLAT scenes and `Truck` by a NERF scene. Top row: Certified pose estimation result. Bottom row: Pose estimation result obtained via sampling. Green: Certified/Correct Camera Regions. Red: Uncertified/Incorrect Camera Regions.

## C.5 ABSTRACTRENDER Results for Scene Variations

In addition to handling camera pose variations as discussed in Section 5, ABSTRACTRENDER can also accommodate scene variations. For GAUSSIANSPLAT scenes, we consider variations in meaningful sets of 3D Gaussian parameters—such as color, mean position, and opacity of objects like trees in a street scene. For NERF scenes, we consider variations in hue or saturation across all 3D point colors in the scene. Table 6 presents ABSTRACTRENDER results under these scene variations for both GAUSSIANSPLAT and NERF representations. Figure 10 visualizes the abstract image of the `PineTree` (a GAUSSIANSPLAT scene) scene under uncertainty in color, mean position, and opacity of two tree objects. Figure 11 shows the abstract image of the `Lego` scene (a NERF scene) under uncertainty in hue and saturation across all 3D points.

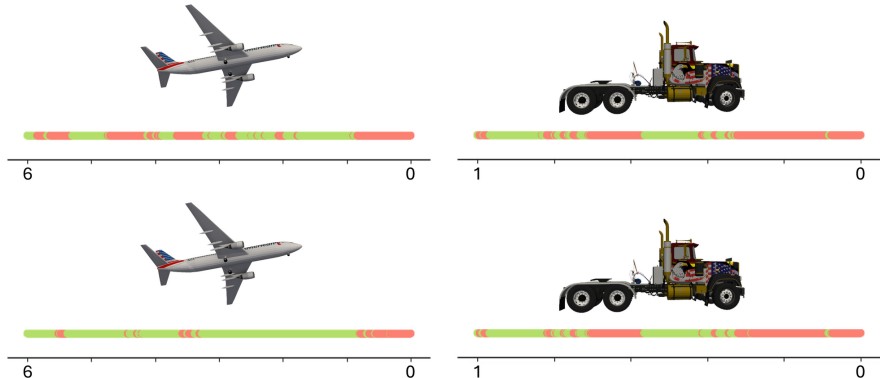

Figure 9: Object detection certification results for `Airplane` and `Truck` in GAUSSIANSPLAT scenes. Top row: Certified object detection result. Bottom row: Object detection result obtained via sampling. Green: Certified/Correct Camera Regions. Red: Uncertified/Incorrect Camera Regions.

| Scene | PR | Dim | SR | Res | Ours | | Empirical | |
|---|---|---|---|---|---|---|---|---|
| | | | | | MPG | XPG | MPG | XPG |
| Lego | 0.15 | hue | NeRF | 80×80 | 0.17 | 1.51 | 0.14 | 1.45 |
| Lego | 0.30 | satur | NeRF | 80×80 | 0.10 | 1.34 | 0.09 | 1.33 |
| Chair | 0.20 | hue | NeRF | 80×80 | 0.65 | 1.71 | 0.37 | 1.71 |
| Chair | 0.50 | satur | NeRF | 80×80 | 0.76 | 1.73 | 0.52 | 1.73 |
| Drums | 0.20 | hue | NeRF | 80×80 | 0.63 | 1.20 | 0.62 | 1.20 |
| Drums | 0.50 | satur | NeRF | 80×80 | 0.96 | 1.36 | 0.88 | 1.36 |
| PineTree | 0.10 | mean | GS | 96×96 | 0.27 | 1.62 | 0.11 | 1.27 |
| PineTree | 0.20 | mean | GS | 96×96 | 0.41 | 1.73 | 0.25 | 1.43 |
| PineTree | 0.10 | op | GS | 96×96 | 0.29 | 1.53 | 0.14 | 1.25 |
| PineTree | 0.20 | op | GS | 96×96 | 0.59 | 1.73 | 0.27 | 1.23 |

Table 6: ABSTRACTRENDER results under scene variations, along with empirical bounds. PR: Perturbation Range; Dim: Perturbation Dimension, satur for saturation, mean for the mean's position of a set of Gaussians, op for opacity of a set of Gaussians; SR: Scene Representation; Res: Rendered Image Resolution; Rt: Runtime (min); MPG: mean pixel gap; XPG: maximum pixel gap.

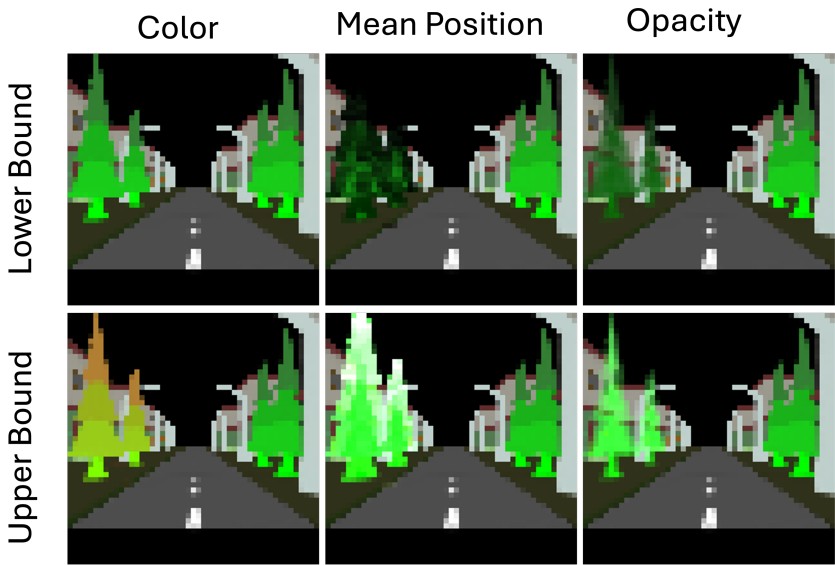

Figure 10: Lower (Top) and upper (Bottom) bound under Gaussian color (Left), mean (Mid) and opacity (Right) perturbation.

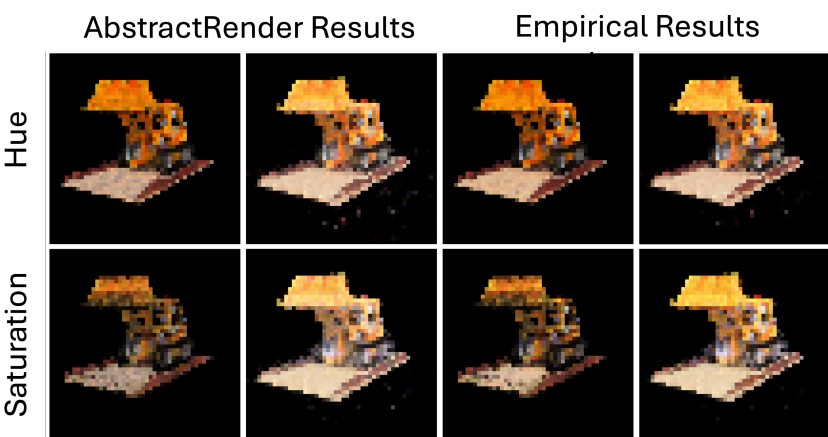

Figure 11: Comparison between ABSTRACTRENDER results and empirical results on `Legot` scene under perturbations in the hue of 3D points (top row) and saturation of 3D points (bottom row).

