# OpenReview forum: "Abstract Rendering:  Certified Rendering Under 3D Semantic Uncertainty"
_NeurIPS.cc/2025/Conference — NeurIPS 2025 spotlight_

### Official Review · Reviewer_ukG3 · 2025-06-11

**Clarity:** 3
**Significance:** 3
**Originality:** 3
**Rating:** 5
**Confidence:** 3

**Summary:**

This paper proposed Abstract Rendering, which is a framework that abstracts NeRF or 3DGS as a collection of operations, where most operations are linearly over-approximable and can be provably bounded. The paper also designs techniques to handle nonlinear operations (Matrix inverse, sort, cumulative product) by designing equivalent boundable algorithms. The paper verifies the soundness of the framework by comparing the results with empirical sampling, and further conducts experiments to demonstrate various applications of this certification framework.

**Questions:**

1. Can you explain whether the framework is dependent on a certain certification framework (CROWN) or can be generalized?
2. Can you refine your examples in Sec. 4 by adding more explanations?
3. Can you compare the uncertainty obtained by the certified bounds with Bayesian uncertainty from a probabilistic perspective, and, if possible, provide experimental comparisons?
4. In Fig. 4, it is surprising that the results of the car and the truck are asymmetric. Intuitively, the result should be symmetric about the axis of $\frac{3\pi}{2}$.
5. In Fig. 5, I can't infer anything from simply looking at the red-green axis. Can you provide some explanations about why this result makes sense?

**Ethical Concerns:**

["NO or VERY MINOR ethics concerns only"]

**Final Justification:**

The rebuttal has addressed most of my concerns, I'll raise my rating toward acceptance.

**Limitations:**

yes

**Quality:**

3

**Strengths And Weaknesses:**

# Strengths
The proposed method is generally novel. It is the first work that interprets the NeRF and 3DGS algorithms using basic operations, provides provable bounds for them, and correctly handles nonlinear operations by further designing tailored techniques to bound them. The soundness of the proposed method is well-validated by experiments, and application-side experiments are conducted to demonstrate the potential of the proposed method.

# Weaknesses
1. Dependent on existing frameworks: After decomposing the rendering algorithm into operations, it requires an existing framework (CROWN) to provide the certification, which is not the paper's contribution.
2. Possible refinements in the examples in Sec. 4: In Example 1 in Sec. 4.1, the original matrix $X$ is not given; is this a mistake? In Example 2 and Fig. 2, it is unclear how the mid-left image's ("via GaussianSplat") upper bound is generated, since the "Sort" operation in GaussianSplat is not linearly over-approximable.
3. Missing comparisons: The experimental results look good, but there is no comparison or verification for these results. A possible line of work that is worth comparing to is the Bayesian uncertainty quantification in NeRFs (e.g., [1, 2, 3]). How does the uncertainty obtained by the certified bounds (via CROWN) compare to the Bayesian uncertainty? Are they similar or different in purpose or applications?

```
[1] Goli, Lily, et al. "Bayes' rays: Uncertainty quantification for neural radiance fields." Proceedings of the IEEE/CVF Conference on Computer Vision and Pattern Recognition. 2024.
[2] Jiang, Wen, Boshu Lei, and Kostas Daniilidis. "Fisherrf: Active view selection and mapping with radiance fields using fisher information." European Conference on Computer Vision. Cham: Springer Nature Switzerland, 2024.
[3] Lyu, Linjie, et al. "Manifold Sampling for Differentiable Uncertainty in Radiance Fields." SIGGRAPH Asia 2024 Conference Papers. 2024.
```

---

> ### Author Rebuttal · Authors · 2025-07-31
>
> We thank the reviewer for recognizing the novelty of our approach in decomposing neural rendering pipelines into basic operations and computing provable bounds, as well as for highlighting our tailored techniques for bounding nonlinear components. We emphasize that this contribution is both technically nontrivial and foundational, as it enables a new class of formal verification and certification tools for vision-based autonomous systems in safety-critical settings. Below, we address the reviewer’s three primary concerns:
>
> **W1**: Dependence on existing verification tool, CROWN.
>
> **W2**: Unclear explanation of examples in Section 4.
>
> **W3**: Lack of experimental comparison; Bayesian uncertainty quantification in NeRFs is suggested as a baseline.
>
> ## **Response to W1** — Our work is built on CROWN, which is not our contribution. :
>
> Our abstract rendering method is implemented using CROWN but, importantly, it does not depend on CROWN specifically. Any tool or library for computing and propagating piece-wise linear bounds can be used to implement our approach. For example, tools like AI^2[1], ERAN[2], convex_adversarial[3] could in principle be used to implement abstract rendering provided they support the operations in Table 4 in the appendix. That said, we note that we *did not* have to extend CROWN with 1) providing linear bound for common operations and 2) propagating linear bound operation by operation for our current implementation. The contribution of our work is in recognizing how the nonlinear functions involved in rendering Gsplat and NeRF can be soundly and compositionally approximated so that the scheme can then be implemented using a library like CROWN.
>
> We are failing to see why using an existing framework (CROWN) for building abstract rendering is a weakness of our work. In the updated paper, we will make the exposition clearer on (a) the current dependence on CROWN and (b) the implementation work needed to build AR on a different framework.
>
> ## **Response to W2** — Confusions on Example 1&2 in Section 4:
>
> We acknowledge that the explanation of the examples in Section 4, was insufficiently detailed due to space limitations. We appreciate the opportunity to clarify the settings and rationale behind our design.
>
> In Example 1, the matrix X is not explicitly provided because the intention is to illustrate how to compute bounds on the inverse of a matrix over *a range of input*, rather than a single instance. While the MatrixInv computes bounds for a fixed matrix X relative to a reference matrix X0, in practical verification pipeline, we work on bounding the inverse over a set of possible matrices, described by element-wise lower and upper bound $\underline{X}$ and $\overline{X}$, respectively.
>
> Given that all operations within MatrixInv support linear bound propagation (as enabled by CROWN), we propagate bound through those operations and obtain element-wise lower and upper bounds of lXinv and uXinv. Then the lower bound of lXinv and upper bound of uXinv form the guaranteed bounds for $X^{-1}$.
>
> The further example is provided in case you are not familiar with linear bound propagation. To illustrate the general approach of this technology, consider a simple function:
>
> $f(x)=(ReLU(x))^2, for x \in [-1,1]$
>
> We decompose this into two functions:  h(x)=ReLU(x) and g(y)=y^2. For h(x), we can linearly bound the output within:
>
> $0\leq h(x)\leq x/2+1/2, for all x \in [-1,1]$
>
> Given $y=h(x)\in [0,1]$, we then bound g(y) with:
>
> $0\leq g(y)\leq y$
>
> By composing the two bounds, we obtain a valid linear bound for f(x) over the range $[-1,1]$:
>
> $0\leq f(x)\leq x/2+1/2$
>
> The resulting interval for f(x) is therefore $[0,1]$.
>
> This example demonstrates the core idea of **linear relaxation-based bound propagation**, which is central to our method. For a complete and rigorous treatment of how linear bounds are computed and propagated across various operations, we refer you to the CROWN series of papers.
>
>
> In Example2, we acknowledge that the baseline approach for generating the mid-left image of Figure 2 (“via GaussianSplat”) is not explicitly explained. It’s true that the *Sort* operation is not supported by linear approximation. To handle this, we concretize all linear bounds into interval bounds by computing their minimum and maximum values over the input domain before the *Sort* operation. The rest of the computation is then conducted using interval bounds. We will clearly explain this in the revision. This transformation inherently loses precision, as interval bounds are typically looser than their linear counterparts. We provide a simple, numerical example in response to W3 of reviewer n9Gh for better illustration..
>
> ## **Response to W3** — Missing comparison with potential baselines (e.g. Bayesian uncertainty quantification in NeRF). :
>
> Thank you for highlighting related work on Bayesian uncertainty quantification in NeRFs . While these approaches are valuable, they address a fundamentally different problem and are not directly comparable to our setting. We clarify the distinction below.
>
> Bayesian uncertainty methods aim to estimate confidence at the *3D point level* in the scene or at the *pixel-color level* in rendered images. Their goal is to indicate which parts of the NeRF reconstruction are reliable or uncertain, which is useful for assessing reconstruction quality or guiding further training.
>
> In contrast, our framework—*abstract rendering*—operates under the assumption that the reconstructed 3D scene is fixed and reasonably represents the underlying real-world structure. We compute an *abstract image set* that is guaranteed to enclose all images rendered from a specified region of camera poses. This set is used to provide *certified input bounds* to downstream ML models (e.g., classifiers or pose estimators), enabling *provable guarantees* on the model’s output across the entire range of physically realizable viewpoints.
>
> Because the Bayesian approach provides **point-wise or pixel-wise confidence** based on **scene reconstruction-relevent camera poses**, while our method yields a **set-valued output** over **a range of camera poses covering the operational domain where the downstream ML model is deployed**, the two outputs are inherently different in form and purpose. As such, a direct comparison is not meaningful.
>
> That said, Bayesian uncertainty quantification is still relevant to our setting, as it informs the reliability of the underlying scene reconstruction—an assumption on which our framework relies. We will add the following statement to the discussion section of the paper:
>
> Our framework assumes the reconstructed 3D scene accurately reflects the real-world environment. Although all model-based analyses involve approximation, understanding the uncertainty in 3D scene reconstruction (e.g., [1, 2, 3]) is meaningful. Incorporating such uncertainty into our framework is a promising direction for future research.
>
>
> ## **Response to Q1** — Please explain whether our framework is dependent CROWN or can be generalized:
>
> As we discussed in the response to W1, our framework can be generalized to other verification tools, as long as they support linear bound propagation among all operations listed in Table 4 in the appendix, except for the three custom operations in our main paper.
>
> ## **Response to Q2** — Please provide more explanations on examples in Sec. 4. :
>
> Yes, thanks for pointing out the insufficient explanation. Please read our response to W2. We will put these explanations in the appendix in the final version.
>
>
> ## **Response to Q3** — Is it reasonable to compare our results with the uncertainty obtained by the certified bounds with Bayesian uncertainty? If so, provide some experimental comparisons.:
>
> Please see our response to W3.
>
> ## **Response to Q4** — Why are the results of the car and the truck in Fig 4 asymmetric? :
>
> The observed asymmetry primarily arises from the behavior of the downstream machine learning models (e.g., the image classifier or pose estimator). Although the objects themselves may exhibit geometric symmetry, these downstream models are not inherently equivariant to symmetric transformations in the input space. As a result, symmetric input images can lead to different outputs.
>
> ## **Response to Q5** — Please provide some explanations on Fig 5 about why this result makes sense. :
>
>
> Figure 5 visualizes which subregions within a continuous 3D camera pose interval can be formally verified to ensure the downstream pose estimator (GateNet) maintains a bounded error. We model the full pipeline—rendering (Gaussian Splatting or NeRF) followed by pose estimation—and apply linear bound propagation using abstract rendering and CROWN.
> Each camera subregion is marked *green* if the upper bound of the pose error remains below a predefined safety threshold (e.g., 10 feet per axis); otherwise, it is marked *red*, indicating unverifiable or potentially unsafe predictions.
>
> Although difficult to fully interpret black-box model behavior, we observe consistent patterns in unverifiable (red) regions, which often correspond to: 1) Unfavorable viewpoints, 2) Occlusion or dominant object features (e.g., a large logo), or 3) Insufficient visual information (e.g., only part of the object visible).
>
> These results highlight the importance of viewpoint diversity and geometric coverage—factors our verification method captures effectively.
>
>
>  ## **Reference**
>
> [1] Gehr, Timon, et al. "Ai2: Safety and robustness certification of neural networks with abstract interpretation." 2018 IEEE symposium on security and privacy (SP). IEEE, 2018.
>
> [2] Müller, Mark Niklas, et al. "ERAN: ETH robustness analyzer for neural networks." URL https://github. com/eth-sri/eran (2022).
>
> [3] Wong, Eric, and Zico Kolter. "Provable defenses against adversarial examples via the convex outer adversarial polytope." International conference on machine learning. PMLR, 2018.

---

> > ### Comment · Reviewer_ukG3 · 2025-08-03
> >
> > Thanks for your detailed response. My questions and concerns are mostly addressed in your rebuttal, while I have some remaining minor questions below:
> > 1. (Relevant to Q5) Your explanation of the meaning of the axis seems reasonable. However, it would be good to explain the meaning of the axis and coordinates in Fig. 5, on how it corresponds to a specific pose, since the camera pose is in 3D, having lots of degrees of freedom. Do you use a single rotational camera model centered on the object with a fixed radius, where the axis represents the rotation angle?
> > 2. (Relevant to W1/Q1, optional) In your rebuttal, you claimed that your method is framework independent, as long as the framework supports operations in Table 4 in the appendix, and provides several examples of other frameworks. It would be nice to support this by conducting some experiments (at least toy experiments) on other frameworks than CROWN.
> >
> > I maintain a positive attitude towards this paper, but providing more explanations and results can further strengthen this paper.

---

> > > ### Author Response · Authors · 2025-08-08
> > >
> > > Thanks for your feedback and we are glad to address your following up concerns.
> > >
> > > ### **1 Explanation of the axis and camera movement coordinates in Fig.5**
> > > We acknowledge that Fig. 5 can benefit from clearer explanation. In our visualization, each axis represents a linear range of camera translations, while the camera is always oriented toward the object center. Unlike the result in Fig 4, here the camera does not follow a fixed-radius circular path; the distance to the object varies along the path. For example, for the plane-GS, the camera moves 6 meters along the airplane’s fuselage, starting from a point that is shifted (2, 2, 1) meters from the object’s center. For the truck-GS, the camera path is 1 meter long along the  truck’s length, starting from a point shifted (0.5, 0.5, 0.5) meters from the object’s center.  For the truck-NeRF cases, the line spans 2 meters and is not parallel to the truck body direction, starting from a point  1m right of the object center. We will clarify these details in the main paper.
> > >
> > > ### **2 Toy experiment of AR on other tool than CROWN**
> > >
> > > We tested ERAN, a neural network verification tool, and found that it supports most basic operations in our pipeline: *Add*, *Mul*, *Mmul*, *Transpose*, and *Exp*. Some additional operations—such as *Div*, *Pow*, *Sum*, *Prod*, and *Norm*—can be constructed from these basic ones. However, operation *Ind* is not supported. We successfully implemented Algorithm 2 (*MatrixInv*) within ERAN, but could not implement Algorithm 3 (*VR-Ind*) due to the lack of support for *Ind*.

---

### Official Review · Reviewer_CPny · 2025-06-27

**Clarity:** 3
**Significance:** 2
**Originality:** 3
**Rating:** 4
**Confidence:** 3

**Summary:**

This paper proposes a verification framework that computes provable bounds on all images rendered by neural rendering methods such as NeRF and 3DGS under perturbed camera poses. This is achieved through three approximations of non-linear operations, such as matrix inversion and sorting.

**Questions:**

- Could the authors clarify in what practical scenarios that the verification of 3D rendering would be effective?
- Can the authors provide more detailed results for real-world scenes?
- Can it be possible to provide several results where all settings are kept the same, except for the rendering method (NeRF vs. 3DGS)?

**Ethical Concerns:**

["NO or VERY MINOR ethics concerns only"]

**Final Justification:**

After the rebuttal, many of my concerns were addressed. Although some points remain unclear regarding practicality, I understand that abstract rendering can be useful in certain contexts. I believe this can facilitate further research, and therefore I raise my rating to 4.

**Limitations:**

yes

**Quality:**

3

**Strengths And Weaknesses:**

### Strength
- The application of formal verification to neural rendering is novel.
- The proposed approximations are methodologically and mathematically sound, and clearly written.

### Weakness
- It is unclear whether the verification, which is essential for assessing the robustness of general neural networks, is also meaningful for 3D rendering models. In rendering, pixel-level differences caused by camera perturbation are expected and do not indicate a failure or lack of robustness. Moreover, the practical use of rendered images from an already trained model for downstream tasks remains unclear.
- Despite using small, object-centric scenes, the verification runtime is reported to take hours for downstream evaluations, raising concerns about scalability.
- The evaluation setup appears inconsistent. For example, comparisons across methods (NeRF vs. 3DGS) are not controlled under the same configuration.
- Adding the perS to the main paper, rather than only the appendix, would help clarify the extent of over-approximation introduced by the method.

---

> ### Author Rebuttal · Authors · 2025-07-31
>
> We thank the reviewer for recognizing the novelty of applying formal verification to neural rendering and for appreciating the mathematical soundness of our Abstract Rendering (AR) method. We maintain that this contribution is not only technically nontrivial, but also enables a new class of verification and certification tools for safety-critical vision-based autonomous systems. Below, we address the reviewer’s two primary concerns:
>
> **W1**: The reviewer questions the practical value of verifying 3D rendering models, noting that (1) pixel-level differences caused by camera perturbations are expected and may not indicate failure, and (2) it is unclear whether verifying rendered images from a pre-trained model is useful for downstream tasks.
>
> **W2**: Verification takes a long time and may not be practical.
>
> We agree that W1 and W2 are important considerations for real-world deployment and end-to-end certification. However, we respectfully maintain that these concerns *do not undermine the central technical contributions of the paper*.
>
> ## **W1**：
>
> We would like to clarify that pixel-level variation due to camera pose changes can in fact lead to significant downstream effects in safety-critical systems such as autonomous driving and robotic manipulation. Seemingly minor rendering differences can alter object detection outcomes or control decisions, particularly when models are sensitive to input shifts.
>
> Our Abstract Rendering (AR) method does not aim to eliminate image variation under camera pose shifts; rather, it allows us to verify that a downstream perception module behaves **consistently and safely** across the set of rendered images bounded by camera uncertainty.
>
> We demonstrate this explicitly in Section 5 through two end-to-end pipelines: 1) AR + image classifier; 2) AR + pose estimator. These experiments show that Abstract Rendering can support **concrete downstream verification tasks**, not just pixel-level reasoning.
>
> As another example, we are currently collaborating with a major aerospace company to apply AR to the **certification of autonomous formation flight** and **landing systems**. In formation flight, a camera on the follower aircraft captures images of the leader. A vision-based ML model estimates the leader’s relative pose, which is used by the flight controller to maintain formation. The company’s goal is to formally certify that, across all feasible relative positions—and under modest environmental variation (e.g., lighting)—the vision system produces pose estimates within a specified error bound.
>
> In this use case, the certification is carried out on a fixed reconstructed 3DGS scene of the environment—e.g., with clear blue skies. (Our framework also handles scene variations with results provided in the table at the end of the response to reviewer n9Gh.) For the formation flight problem, being able to certify the system under nominal conditions (e.g., fixed lighting, background landscape, etc.) is already considered an important advance, as it ensures that even in idealized aerospace deployments, the vision-based controller behaves reliably and predictably.
>
>
> ## **W2**:
>
> We appreciate the reviewer’s concern regarding verification runtime. **Provable, worst-case verification is inherently more computationally demanding than empirical testing**, as it guarantees correctness over entire input regions—not just sampled instances. In this regard, AR is aligned with the broader formal methods community, where high runtime is an accepted tradeoff for soundness.
>
> For comparison, leading neural network verification methods such as **β-CROWN** [1], **NeuralSAT** [2], and **PRIMA** [3] routinely require **hundreds to thousands of seconds per instance**. Our reported runtime—ranging from **minutes to a few hours** depending on scene complexity and perturbation bounds—is well within the standard range for safety verification tasks. This is particularly reasonable given that our work is the **first to provide provable bounds through differentiable 3D rendering models** such as NeRFs and 3DGS.
> We emphasize that **our current implementation prioritizes tightness and correctness of bounds**. As in the early stages of many verification tools (e.g., CROWN [4]), we begin with small, controlled settings to develop foundational techniques. Over time, tools like CROWN scaled up to handle larger networks and more complex tasks—culminating in competitive results on VGG-16 and ResNet benchmarks in VNN-Comp 2024. We are following a similar path and are actively developing optimizations, including: 1) Parallelization of bound computation, 2) GPU acceleration for abstraction steps, and 3) Tighter approximations for nonlinear operations.
>
> ## **W3 & Q3**
>
> We agree that using consistent configurations is essential when comparing a novel method against a baseline, as it ensures fairness and highlights the true contribution of the proposed work. However, in this case, **both abstract rendering pipelines over NeRF and 3DGS are contributions of our own framework**. Therefore, a direct performance comparison between the two is not the **primary objective** of our experiments setup.
>
> Instead, the purpose of our experiment setup (e.g. Table 1) is to demonstrate that our verification framework is compatible with multiple rendering backbones and can operate under a variety of configurations. Given the page limit, we aimed to showcase this breadth rather than perform controlled comparisons.
>
> Although our goal is not to directly compare the verification pipeline on NeRF and 3DGS, given your interest in a shared configuration, we updated Table 1 to include both rendering methods under the same setup, as shown below.
>
> |Scene|ε|Dim|Res|**GS**|||**Nerf**|||**Empirical**||
> |---|---|---|---|---|---|---|---|---|---|---|---|
> ||| | |Rt|MPG|XPG|Rt|MPG|XPG|MPG|XPG|
> |Lego|0.1(rad)|yaw|80×80|22|0.51|1.73|25|0.40|1.73|0.22|1.57|
> |Chair|0.1(rad)|yaw|50×50|17|0.46|1.73|20|0.40|1.73|0.40|1.72|
> |Drums|0.1(m)|x|50×50|5.3|0.19|1.62|5.8|0.16|1.55|0.13|1.54|
> |PineTree|2(m)|x|72×72|4.1|0.27|1.73|4.9|0.23|1.73|0.06|1.37|
> |PineTree|10(m)|x|72×72|24|0.47|1.73|28|0.36|1.73|0.18|1.37|
> |Airport|0.027(rad)|roll|160×160|27|0.56|1.69|35|0.42|1.63|0.21|1.38|
> |Garden|0.5(m)|x|100×100|20|0.53|1.59|11|0.37|1.53|0.34|1.21|
>
>
> ## **W4**
>
> *perS* will be moved to the main paper in the final version.
>
>
> ## **Q1**
>
> Our verification framework is particularly well-suited to **vision-based, safety-critical applications** where the environment is structured, the number of relevant objects is limited, and high reliability is essential. Examples include:
>
> **Autonomous formation flight** (as discussed above), where a follower aircraft must maintain precise positioning relative to a lead aircraft using camera input.
>
>
> **Vision-based automated landing**, where a drone or aircraft must visually identify and align with a landing site based on onboard perception.
>
>
>
> **Autonomous vehicle docking or payload transfer**, where AR ensures that visual pose estimators remain accurate across approach angles during close-range maneuvers.
>
>
> **Warehouse inventory robotics**, where AR verifies that a robot navigating structured aisles can reliably detect and classify labeled bins or packages from varying camera angles under fixed lighting.
>
>
> In all of these scenarios, **modest environmental variability** (e.g., fixed lighting, sky backgrounds, known landmarks) makes it feasible to reconstruct a high-fidelity 3D scene (e.g., via 3DGS), and it is crucial to certify perception behavior over a **bounded set of viewpoints**. AR enables **provable safety guarantees** in these settings—where failures can have high consequences and formal correctness is preferred over empirical coverage.
>
> ## **Q2**
> We provide additional results on more realistic scenes featuring large, complex backgrounds. The evaluation includes four scenes: Garden and Bicycle from the Mip-NeRF 360 dataset, as well as Airport and Airplane with Mountain View Background from our own benchmark. These new scenes are chosen to test the robustness of our framework under varied and realistic conditions.
>
> |Scene|ε|Dim|res|**GS**|||**NeRF**|||**Empirical**||
> |------------|-----------|-------|-------------|--------------|---------|---------|-------------|---------|---------|--------------|---------|
> ||| | |Rt|MPG|XPG|Rt|MPG|XPG|MPG|XPG|
> |Garden|0.5(m)|x|100×100|20|0.53|1.59|33|0.37|1.53|0.34|1.21|
> |Bicycle|0.2(m)|x|100×100|23|0.72|1.68|31|0.58|1.60|0.37|1.35|
> |Airport|0.03(rad)|roll|160×160|30|0.59|1.70|43|0.51|1.68|0.22|1.41|
> |Airplane|2(m)|x|80×80|16|0.41|1.68|19|0.33|1.61|0.15|1.39|
>
>
> ## **Reference**
>
> [1] Wang, Shiqi, et al. "Beta-crown: Efficient bound propagation with per-neuron split constraints for neural network robustness verification." Advances in neural information processing systems 34 (2021): 29909-29921.
>
> [2] Duong, Hai, ThanhVu Nguyen, and Matthew Dwyer. "A dpll (t) framework for verifying deep neural networks." arXiv preprint arXiv:2307.10266 (2023).
>
> [3] Müller, Mark Niklas, et al. "PRIMA: general and precise neural network certification via scalable convex hull approximations." Proceedings of the ACM on Programming Languages 6.POPL (2022): 1-33.
>
> [4] Zhang, Huan, et al. "Efficient neural network robustness certification with general activation functions." Advances in neural information processing systems 31 (2018).

---

> > ### Comment · Reviewer_CPny · 2025-08-05
> >
> > Thank you for the explanation.
> > As I reviewed, I acknowledge the novelty of this work. To my knowledge, Abstract Rendering is introduced for the first time in this field with solid grounding, and several of my earlier concerns have been addressed in the rebuttal.
> >
> > However, I still have remaining concerns regarding the practicality of the approach (W1), which was also pointed out in Reviewer fGys’s official comment 1.
> > I believe a clearer articulation of the practical applicability is necessary for AR to become impactful in the way NN verification has been—serving as a foundation for future developments and improvements.
> >
> > The formation flight use case is interesting, but I am still unclear on why AR is needed in this context. Could you elaborate further?
> >
> > As I understand it, the pipeline involves first training a 3D model with a nominal background scene, and then using this model in conjunction with a downstream model (pose estimator) to perform certification by predicting error bounds.
> > If this is correct, what would be the next step after such certification?
> > If my understanding is incorrect, could you provide a more detailed explanation of how AR contributes to the overall pipeline?

---

> ### Author Response · Authors · 2025-08-08
>
> Thank you for your thoughtful follow-up and for recognizing the novelty and technical grounding of Abstract Rendering (AR). We appreciate your continued engagement and the opportunity to clarify the practical role of AR, particularly in the formation flight example.
>
> Kindly see our response to Reviewer fGys, where we address broader concerns about the practicality of AR, provide new experimental results, and clarify how modest scene variations are handled (e.g., S2). Here, we elaborate specifically on how AR fits into the certification pipeline for visual autonomy.
>
> Your understanding of the formation flight use case is broadly accurate. The pipeline involves:
>
>   - **Scene reconstruction**: We assume a high-fidelity 3DGS model of the scene including its variations. (This step is valuable even beyond verification—for example, to train perception or control policies.) In the S2 experiments, to model simple background variation, we generate a foreground 3DGS for the airplane and interpolate the background separately. This construction of the 3DGS is tailored to the specific variation being studied.
>
>
>
>   - **Uncertainty propagation via AR**: AR propagates camera pose and scene uncertainty through the 3DGS model to generate an abstract image—a conservative over-approximation of all rendered images that could arise under bounded pose and scene perturbations.
>
>   - **Downstream verification**: This abstract image is then passed to a neural verification tool to certify that the downstream classifier or pose estimator remains within acceptable error bounds across the entire range of camera motions (and scene variations).
>
> In the context of formation flight, AR enables formal end-to-end reasoning about the vision pipeline. It verifies that, under all feasible approach angles and modest scene variations, the downstream system behaves predictably. The bounds produced by AR define a *region of guaranteed correct behavior over the space of (camera and scene) variations*.
>
>
> ### **What happens after certification?**
>
> Successfully verifying the pipeline produces certification artifacts for the system and the ML models—the reconstructed scene, region of correct behavior (often called the operating design domain or ODD), and verification results. These artifacts can be used as part of a broader safety case—e.g., internal within an organization or for FAA or EASA—just as test results and formal proofs are used today in DO-178C workflows for airborne systems.
>
> Beyond certification, AR supports **design iteration**:
>
>   - **If verification fails**, the counterexample region highlights scenarios where perception is fragile, guiding data augmentation or retraining. See exemplary results below.
>
>   - **If verification is inconclusive**, it flags regions requiring tighter abstractions or further analysis.
>
>
>
>   - **If verification succeeds**, it provides *provable* guarantees over a space of structured inputs—something empirical testing cannot offer.
>
> Thus, AR is both a certification tool and a development aid, similar to how SMT solvers (e.g., CBMC) are used in safety-critical software. While modest in scope today, AR introduces a new formal capability that can scale with increasing regulatory demands on vision-based autonomy.

---

### Official Review · Reviewer_n9Gh · 2025-07-02

**Clarity:** 3
**Significance:** 3
**Originality:** 3
**Rating:** 5
**Confidence:** 3

**Summary:**

The paper first proposes AbstractRendering, an approximate way to estimate the lower and upper bound of the rendered images from a 3D scene given a linear set of camera poses.
The paper studies the math operations used in the 3D-GS and NeRFs, especially the (1) matrix inversion (2) sorting-based summation (3) and cumulative-product-based summation, and proposes better estimation for these operators.
Experiments are conducted to examine the effectiveness of the proposed method.

**Questions:**

- Why there are N/A values for PSNR/SSIM in Table 5 of the updated appendix?

**Ethical Concerns:**

["NO or VERY MINOR ethics concerns only"]

**Limitations:**

yes

**Quality:**

3

**Strengths And Weaknesses:**

Strength:
- The idea of the rendering bound under the noisy camera pose is novel and insightful.
- The paper is well-organized and easy to follow.
- The experiments support the claims well.

Weakness:
I am not an expert in approximation and have not kept up with recent progress in that field. The motivation, theories, and results look reasonable to me. I have only some questions that are not necessarily the weakness of the paper:
- The 3D-GS scene model described in Section 3.2 L.137, directly uses the RGB vector $\mathbf{c}$ to model the color instead of the wide-used SH coefficient. Is there any specific reason that the SH coefficient is not used? To my understanding, similar approximation can be made to estimate the procedure of calculating the color from the SH coefficients.
- In L.159, the linear set of cameras is easy to understand by flatting the camera parameters into one single vector and applying the definition of linear set in L.116 (or by the spatial distribution of the noisy cameras). However, as both Gaussian and NeRF scenes have lots of parameters, how to properly define a linear set of the Scene $\mathbf{Sc}$?
- I am a little confused about Table 1. The paper mentioned in L.282 that *the NeRF can leverage the linear bound relaxation for cumulative product summation* so it shows smaller MPG/XPG than the 3D-GS. Is it the only reason to cause such a difference? Will the intrinsic robustness (to viewpoint changes) of 3D representations affect the MPG/XPG metric? For example, if one representation reconstructs the scene better (e.g. with more dense source views, or with the modeling of viewpoint effect such as the ray direction encoding in NeRF or SH in 3D-GS), will it have a smaller MPG/XGP metric for the same scene, compared to the other 3D representation of the same kind whose reconstruction quality is worse? Furthermore, is there some quantitative comparison between the proposed method and the sampling-based method (as Fig. 3)?

---

> ### Author Rebuttal · Authors · 2025-07-31
>
> We thank the reviewer for their positive evaluation and thoughtful questions. We’re glad to see that the novelty and the empirical support of the paper came through clearly. Below, we address each of the reviewer’s points:
>
> ## **Response to W1** — Why use RGB, not SH to model GS color:
>
> We appreciate the reviewer’s thoughtful question. In our current implementation, we chose to use *fixed RGB vectors* to model color rather than *Spherical Harmonic (SH) coefficients*, primarily to keep the abstraction and analysis tractable for this initial study.
>
> While SH coefficients are indeed widely used in 3DGS to model *view-dependent appearance*, incorporating them would introduce additional *nonlinear dependencies on the viewing direction*—specifically, trigonometric basis functions and dot products over the camera ray and normal vectors. This would make the rendering process *significantly more nonlinear*, increasing the complexity of the abstract interpretation, especially under camera pose uncertainty.
>
> That said, we fully agree that SH coefficients offer *greater expressive power*, allowing splats to vary their appearance with camera angle and enabling the modeling of specularities and shading effects. Our current work lays the foundation for bounding color contributions in photorealistic rendering pipelines, and extending our framework to handle *SH-based view-dependent color* is a natural and important direction for future work. The key techniques introduced here—such as bounding composition of nonlinear functions like matrix inversion and sorting—will generalize to that setting.
>
>
> ##  **Response to W2** — How to properly define a linear set of the Scene Sc?:
>
> In 3DGS, scenes are represented by sets of 3D Gaussians parameterized by color, opacity, mean, and covariance. A naive approach flattens these parameters into a vector for interpolation, similar to camera interpolation, but this becomes impractical due to the large number of Gaussians. We are actively exploring a more efficient method that enables physically meaningful scene variations—such as lighting, background, or coordinated object-specific changes. This is a realistic and relevant challenge. Similar to the formation flying scenario discussed in our response to Weakness 1 of reviewer rGhy, we aim to verify whether the target pose estimator remains robust under varying lighting and background conditions. Existing results for some type of scene variations—hue and saturation for NeRF, and brightness and background scenarios for GS, are provided in the table at the end of this rebuttal.
>
>
> ## **Response to W3** — What causes lower MPG/XPG for abstract images from NeRF compared to 3DGS? Is it solely due to the use of linear bound relaxation for cumulative product summation? :
>
> The difference in MPG/XPG between NeRF and 3D-GS is primarily due to the former’s ability to leverage linear bound relaxation over cumulative product summation. However, this is not the only contributing factor. Other aspects such as the density of source views and the inclusion of viewpoint modeling (e.g., ray direction encoding in NeRF or spherical harmonics in 3D-GS) can also influence the bounds.
>
> In our experiments, we control for these variables by using the same source views and training settings for both methods. Even when 3D-GS achieves better rendering quality than NeRF, it consistently produces looser pixel-level bounds under camera rotation, due to the inapplicability  for linear bound propagation through the depth-based sorting operation. This is because, in a typical 3D-GS scene (with ~50k Gaussians), 500–6000 Gaussians can contribute significantly to a single pixel, even after applying opacity and distance thresholds. Due to the cumulative product over depth ordering, small uncertainties in depth can easily disrupt the correct ordering, leading to substantial over-approximation for provable (worst-case) bound.
>
> In contrast, NeRF samples a fixed, small number of points (typically 32–128) along each ray, with a clearly defined front-to-back order, enabling linear bound propagation and resulting in much tighter bounds. The visual difference in bound tightness of a simple scene containing only three Gaussians is illustrated in the mid-left (naïve interval bound) and mid-right (linear bound) images in Figure 3. An additional numerical example is provided below for further clarification.
>
> ### **Illustrative Example:**
>
> Let us consider the following setup:
>
> 1)    Scalars: $a={0.8,0.5}$, $d={x,x+1}$, where $x\in [-1,1]$,
>
> 2)    Goal: Compute a tight and worst-case bound of the following procedure:
>
> Step1: $as=sort(a,d)$;
>
> Step2: $oc[0]=1$;
>
> Step3: $oc[1]=1-as[0]$;
>
> Step4: $pc=oc[0]\times as[0]+oc[1]\times as[1]$;
>
>
> *Naïve interval approach*:
>
> Because sort(a,d) depends on the uncertain ordering of $d[0]=x\in [-1,1]$, and $d[1]=x+1\in [0,2]$, we can not determine the correct ordering of a[0] and a[1] in *as*. Thus we must conservatively assume the worst case:
>
> $as[0], as[1] \in [0.5,0.8]$;
>
> $oc[1] \in [0.2,0.5]$.
>
> The resulting interval bound for pc is:
>
> $pc\in [1⋅0.5+0.2⋅0.5,1⋅0.8+0.5⋅0.8]= [0.6,1.2]$
>
> This yields a very loose bound.
>
> *Proposed VR-Ind approach*:
>
> Using our proposed method *VR-Ind*, the same computation is encoded symbolically to account for value-dependent decisions. The control flow is represented using indicator functions (*Ind*), allowing us to keep the dependencies on *d* explicit:
>
> Step 1: $oc[0]=(1-a[0]\times Ind(d[0]-d[0]))\times  (1-a[1]\times Ind(d[0]-d[1]))$
>
> Step 2: $oc[1]= (1-a[0]\times Ind(d[1]-d[0]))\times  (1-a[1]\times Ind(d[1]-d[1]))$
>
> Step 3: $pc=oc[0]\times a[0]+oc[1]\times a[1]$;
>
> Given that
>
> $d[0]-d[1]=x-(x+1)=-1<0 \rightarrow Ind(-1)=0$;
>
> $d[1]-d[0]=(x+1)-x=1>0 \rightarrow Ind(1)=1$;
>
> $Ind(0)=0$;
>
> We obtain that
>
> $oc[0]=(1-0.8\times 0)(1-0.5\times 0)=1$;
>
> $oc[1]=( 1-0.8\times 1)(1-0.5\times 0)=1-a[0]=0.2$;
>
> $pc= 1\times 0.8+0.2\times 0.5=0.9$.
>
> Thus, we obtain a precise bound $pc\in [0.9,0.9]$, which is significantly tighter than the interval-based bound of $[0.6,1.2]$. By the way, in the real verification pipeline, VR-Ind will produce a linear relaxation set instead of a fixed value. Here we consider a simple case for highlighting VR-Ind’s advantage in terms of bound tightness.
>
> The naïve interval method (used for mid-left image in Figure 2) leads to loose bounds due to loss of dependency information through Sort operation. In contrast, the proposed VR-Ind method (used for the mid-right image of Figure 2) preserves input-dependent flow and yields significantly tighter bounds.
>
>
> ## **Response to Q** – Why are there N/A values for PSNR/SSIM in Table 5 of the updated appendix? :
>
> The complete Table 5 in the updated appendix:
>
> |Scene|GAUSS(GS)|PSNR(GS)|SSIM(GS)|LPIPS(GS)|PSNR(NeRF)|SSIM(NeRF)|LPIPS(NeRF)|
> |------|---------|--------|--------|---------|-----------|-----------|------------|
> |Lego|43,543|25.60|0.95|0.07|21.38|0.81|0.19|
> |Chair|46,108|22.98|0.94|0.08|22.56|0.89|0.11|
> |Drums|50,877|21.19|0.89|0.10|19.46|0.81|0.23|
> |Pinetree|113,368|31.06|0.97|0.06|22.41|0.79|0.20|
> |Airport|617,371|18.83|0.84|0.33|20.83|0.72|0.22|
> |Garden|524,407|18.74|0.37|0.32|22.15|0.80|0.20|
> |Plane|51,316|28.05|0.95|0.11|22.89|0.70|0.26|
> |Truck|47,895|24.70|0.94|0.09|23.53|0.75|0.19|
> |Car|34,699|26.98|0.93|0.10|20.75|0.69|0.27|
>
>
> ## **Additional result on AR under scene variation**:
>
> The table below shows our framework capable of handling scene variation.
>
> |Scene|ε|Dim|Method|res|**Ours**| |**Empirical**| |
> |---|---|---|---|---|---|---|---|---|
> | | | | | |MPG|XPG|MPG|XPG|
> |Lego|0.15|hue|NeRF|80×80|0.17|1.51|0.14|1.45|
> |Lego|0.3|satur|NeRF|80×80|0.10|1.34|0.09|1.33|
> |Chair|0.2|hue|NeRF|80×80|0.65|1.71|0.37|1.71|
> |Chair|0.5|satur|NeRF|80×80|0.76|1.73|0.52|1.73|
> |Drums|0.2|hue|NeRF|80×80|0.63|1.20|0.62|1.20|
> |Drums|0.5|satur|NeRF|80×80|0.96|1.36|0.88|1.36|
> |Street1|0.1|ob-bright|GS|96×96|0.27|1.62|0.11|1.27|
> |Street1|0.2|ob-bright|GS|96×96|0.41|1.73|0.25|1.43|
> |Street2|0.1|ob-bright|GS|96×96|0.29|1.53|0.14|1.25|
> |Street2|0.2|ob-bright|GS|96×96|0.59|1.73|0.27|1.23|
> |Airplane|0.1|bg_sky|GS|160×160|0.49|1.68|0.19|1.52|
> |Airplane|0.1|bg_mt|GS|160×160|0.75|1.73|0.51|1.44|
>
> Caption: Results of Abstract Rendering under various scene perturbations, compared with empirical results. *Scene*: test scene; *Dim*: type of perturbation. *Hue* and *Satur* refer to hue and saturation changes applied to the entire scene. *ob-bright* modifies the brightness of a specific object—two trees in Street1, and one building in Street2. *bg* represents a setting where the airplane (foreground) is trained using GS, and the background is linearly interpolated between standard sky and mountain images. bg\_sky and bg\_mt correspond to background regions closer to sky and mountain, respectively. res: output image resolution. *MPG/XPG*: Mean/Max Pixel-Color Gap, as defined in the main paper. The *Ours* and *Empirical* columns report MPG and XPG for our method and the empirical baseline, respectively.

---

> > ### Comment · Reviewer_n9Gh · 2025-08-06
> >
> > Dear Authors,
> >
> > Thanks for your detailed response. My questions have been addressed.
> >
> > Best Regards,
> > The Reviewer.

---

> > > ### Author Response · Authors · 2025-08-08
> > >
> > > Dear Reviewer,
> > >
> > > Thank you for your time and thoughtful questions. We're glad our response was helpful.
> > >
> > > Best regards,
> > >
> > > The Authors

---

### Official Review · Reviewer_fGys · 2025-07-03

**Clarity:** 3
**Significance:** 2
**Originality:** 3
**Rating:** 4
**Confidence:** 3

**Summary:**

This paper proposes an “abstract rendering” framework for scenes represented by neural radiance fields, aiming to compute conservative bounds that encompass the set of all possible rendered images produced over a continuous range of camera poses. The authors design algorithms for propagating input uncertainty through the rendering pipeline, constructing conservative bounds on the output images for verification.

**Questions:**

1. Please see the weakness section.
2. What is the practical benefit of the proposed method, compared to dense empirical sampling of the rendering pipeline, especially given the real-time performance of modern 3DGS/NeRF?
3. Colors in Algorithm 1 should be view-dependent, i.e. perturbing the camera may yield different colors for the same Gaussian, which is not captured in the proposed algorithm.

**Ethical Concerns:**

["NO or VERY MINOR ethics concerns only"]

**Final Justification:**

The paper takes a first step to extend abstract image computation techniques from mesh-based rendering to Gaussian splatting and NeRF, which are emerging and popular scene representations.
After the rebuttal, I understand the practical values of the proposed method, especially when compared with the sampling-based method.
The proposed method is still useful even without modelling the reconstruction uncertainty or an advanced rendering scheme, especially considering that adapting abstract image computation to the rendering process for Gaussian splats and NeRF is mathematically nontrivial.

**Limitations:**

yes

**Quality:**

3

**Strengths And Weaknesses:**

**Strengths**
1. The paper extends abstract image computation techniques from mesh-based rendering to Gaussian splatting and NeRF, which are emerging and popular scene representations.
2. Adapting abstract image computation to the rendering process for Gaussian splats and NeRF is mathematically nontrivial.
3. Empirical results show that the proposed method can approximate the pixel-color bounds on the rendered images, which are affected by uncertainty in camera poses.


**Weaknesses**
1. The practical value of the proposed verification is questionable, as the method operates on fixed scene representations obtained via inverse rendering pipelines that already assume known camera parameters. In practice, 3D Gaussian splats or neural radiance fields are optimized to be **consistent** with training camera poses; if the camera poses used during scene capture were uncertain, that uncertainty would already affect the scene representation, not just the rendered images. Thus, verifying possible rendered images from a fixed scene under varying cameras does not reflect real-world verification needs.
2. Questionable Need Over Empirical Approaches: Robustness to camera pose or scene perturbations could be empirically evaluated by sampling (as Table 1 illustrated), given that 3DGS and NeRF rendering are already **real-time**. The paper does not convincingly justify the necessity of complex and approximate abstraction over simple sampling for practical applications.
3. The formal bounds produced are over low-level pixel values, whereas safety properties are typically defined at the level of object detection, semantics, or control decisions. The paper does not address end-to-end safety or correctness of downstream tasks.

---

> ### Author Rebuttal · Authors · 2025-07-31
>
> Our Abstract Rendering (AR) is the *first* to provide *provable guarantees* over ML-based image generation and perception pipelines involving modern photorealistic scene representations such as NeRFs and 3D Gaussian Splats (3DGS). We maintain that this technical contribution is not only nontrivial, but also enables a new class of **verification and certification tools for safety-critical vision-based autonomous systems**.
>
> Your review raises thoughtful concerns regarding the practical value of abstract rendering, particularly:
>
> (W1) that uncertainty in the scene reconstruction (e.g., from camera poses during capture) might undermine the validity of verification of 3DGS and downstream image processing,
>
> (W2) that simple empirical sampling may be sufficient in most practical settings, and
>
> (W3) that the paper does not demonstrate end-to-end verification at the semantic or decision level.
>
> We agree that W1 and W2 are important considerations in the broader context of verifying vision-based autonomous systems. However, we respectfully maintain that these concerns **do not undermine the central technical claims and contributions** of the paper. Our work provides a mathematically sound, general-purpose framework for reasoning about uncertainty propagation in photorealistic rendering pipelines—something empirical methods cannot offer. We address W1–W3, Q1-Q3 in detail below.
>
> ## **W1**
>
> As with any model-based verification approach, the guarantees produced by Abstract Rendering (AR) are conditioned on the fidelity of the underlying model. AR provides *formal guarantees about uncertainty propagation* through the rendering and perception pipeline *with respect to a given 3DGS*. Whether the trained 3DGS itself accurately reflects the real-world scene is a separate—but important—consideration, and falls outside the primary technical focus of this paper.
>
> This distinction is standard in verification: for instance, robustness verification of neural networks assumes a fixed input image, even though the image may itself result from a noisy sensing process. Similarly, AR assumes a fixed 3D scene and computes provable abstract images under bounded camera perturbations. This abstraction is both principled and practical—especially given the increasing fidelity of 3DGS reconstructions enabled by modern multi-view capture and recent advances in scene optimization.
>
> As a concrete example, a major aerospace company is collaborating with us to apply AR to the **certification of an autonomous formation flight system** and an **automated landing system**. For formation flight, a camera mounted on the follower aircraft captures images of the lead aircraft. A vision-based ML model estimates the relative pose of the leader, which is then used by the follower’s flight controller to maintain formation. The company seeks to **formally certify** that across *all feasible relative positions* between the aircraft—and under modest environmental variation (e.g., lighting)—the vision system produces pose estimates within a specified error bound.
>
> This verification problem is conceptually identical to the **certified pose estimation** results presented in our paper. AR enables formal guarantees about the output of the pose estimator over the space of rendered images, which can then be integrated into standard system-level verification pipelines. Due to the safety-critical nature of the task, worst-case guarantees are preferred over probabilistic methods, which may be difficult to interpret or insufficient in high-stakes settings.
>
> In this use case, the certification is carried out on a fixed reconstructed 3DGS scene of the environment—e.g., with clear blue skies. (Our framework also handles scene variations with results provided in the table at the end of the response to reviewer n9Gh.) For the formation flight problem, being able to certify the system under nominal conditions (e.g., fixed lighting, background landscape, etc.) is already considered an important advance, as it ensures that even in idealized aerospace deployments, the vision-based controller behaves reliably and predictably.
>
> As to the question of accuracy of 3DGS (which is admittedly not addressed in the current paper) their fidelity can be improved through: 1) Increasing the number and quality of capture viewpoints, 2) Using high-precision cameras, 3) Applying advanced, recently developed variants of 3DGS that improve reconstruction under pose uncertainty.
>
> In view of the above, we respectfully reiterate that our verification setup is *both technically sound and practically motivated*. The ability to *certify downstream models over a range of plausible viewpoints and modest scene variation* is a powerful and timely capability—one we believe AR delivers.
>
>
> ## **W2 & Q2**
> Empirical sampling is a valuable tool for uncovering edge cases—but as Dijkstra famously said, “testing can show the presence of bugs, not their absence.” In contrast, Abstract Rendering (AR) computes conservative over-approximations that enable formal reasoning. For example, AR can prove that, across all approach angles of a follower aircraft toward a leader (see example introduced above), the downstream pose estimator will produce errors within a specified threshold—under the assumption of a given 3DGS scene. A real application of AR is discussed in our response to W1.
>
> Whether such formal guarantees are preferable to extensive empirical testing is context-dependent. In certain domains such as aerospace, chips, or distributed protocols, formal verification is often an essential complement to testing, especially when rare edge cases can cause catastrophic failure. We expect that similar needs will emerge in autonomy and robotics as systems become more capable and more tightly integrated into real-world environments.
>
> It is also well understood that verification incurs a higher computational cost than testing and is not realtime. For example, verifying the robustness of a neural network on a small perturbation range often takes minutes—far longer than inference over regular inputs. Yet, the community has embraced this tradeoff, as evidenced by the large body of work on neural network verification and the strong participation in benchmarks like *VNN-Comp*. The cost is justified by the value of *soundness*.
>
> In this broader context, AR provides a new capability: **it allows verification of perception models not under arbitrary pixel-level perturbations, but under structured, physically grounded changes in viewpoint**. Our work is the first to propagate formal uncertainty bounds through the **rendering process**, across advanced photorealistic scene representations like NeRFs and 3DGS—filling a key gap in existing verification pipelines.
>
> While our current implementation prioritizes correctness over speed, it serves as a foundation. We are actively exploring optimizations to improve scalability, including 1) parallelization, 2) tighter abstractions, and 3) efficient approximation strategies. We believe this makes AR not just a complement to empirical testing, but a foundational step toward verifiable autonomy.
>
>
> ## **W3**
>
> We would like to point out that our work *does* provide end-to-end correctness guarantees by propagating uncertainty in camera pose (and modest scene variation) through both the rendering process and the downstream image processing models—such as classifiers and pose estimators.
>
> The overall verification process proceeds in two stages: 1) Rendering-stage verification: Abstract Rendering (AR) computes provable over-approximations of all images that could be generated from a given 3D scene and a bounded set of camera configurations. 2) Perception-stage verification: These image sets are then passed through neural networks, whose outputs are formally verified using mature tools developed by the neural network verification community.
>
> The second stage has been extensively studied in the literature. State-of-the-art tools such as alpha-beta-CROWN [1], NeuralSAT [2], PRIMA [3], and others from the VNN-Comp competition have demonstrated robust performance across diverse architectures and tasks. Accordingly, our focus in this paper is on the first stage, where there has been *no prior work*: computing formal, structured uncertainty bounds through neural rendering pipelines like NeRFs and 3DGS.
>
> In Section 5 (Figures 4–5, Tables 2–3), we present experimental results on two concrete end-to-end pipelines: AR + image classifier, and AR + pose estimator. These experiments demonstrate the ability of AR to support complete safety verification pipelines, covering both uncertainty in rendered input and robustness of the perception model’s output. While more complex downstream tasks—such as semantic segmentation or object detection—can be considered in future work, the same pipeline structure applies. As long as the downstream model is a neural network, existing verification tools can be used to extend AR to these settings.
>
> Thus, our work represents a key missing component in enabling full end-to-end safety guarantees for vision-based autonomy: the sound propagation of uncertainty from the physical world, through rendering, into perception.
>
> ## **Q3**
>
> Please see our response to W1 of reviewer n9Gh.
>
> ## **Reference**
> [1] Wang, Shiqi, et al. "Beta-crown: Efficient bound propagation with per-neuron split constraints for neural network robustness verification." Advances in neural information processing systems 34 (2021): 29909-29921.
>
> [2] Duong, Hai, ThanhVu Nguyen, and Matthew Dwyer. "A dpll (t) framework for verifying deep neural networks." arXiv preprint arXiv:2307.10266 (2023).
>
> [3] Müller, Mark Niklas, et al. "PRIMA: general and precise neural network certification via scalable convex hull approximations." Proceedings of the ACM on Programming Languages 6.POPL (2022): 1-33.

---

> > ### Comment · Reviewer_fGys · 2025-08-05
> > **Thanks for your rebuttal.**
> >
> > Thanks for the efforts during the rebuttal. My main concerns remain unsolved, and I look forward to further discussions:
> > 1. The method assumes a fixed and accurate 3DGS/NeRF but does not account for reconstruction uncertainty, which undermines real-world applicability. If the scene representation itself is uncertain, the verification results may be of limited practical use. The rebuttal sidesteps this issue by analogy to NN verification, but unlike images, scene reconstructions directly encode physical reality and are sensitive to capture noise. The aerospace use case is interesting but remains anecdotal and not empirically validated in the paper or rebuttal.
> > 2. Current experiments are restricted to simplified setups and insufficient real-world data, leaving unclear whether AR can scale to realistic environments or diverse perception tasks. While the rebuttal argues from principle (“proof > testing”), it does not demonstrate concrete scenarios where empirical sampling fails but AR succeeds. This is particularly concerning since AR itself relies on simplified models and uses empirical sampling to support its soundness in the Method section.
> > 3. Regarding the physically grounded claims, the rendering process itself involves large approximations (e.g. camera matrix simplifications in 3DGS), which raises doubts on whether the formal guarantees meaningfully reflect actual rendering behavior. Moreover, the current formulation omits important physically-based factors such as reflectance, sufficient real-world data, and accurate volumetric rendering models. Existing physically-based approaches (e.g., EVER, 3DGUT) are not considered, leaving AR underdeveloped in terms of rendering fidelity.
> >
> > I will raise my score if these concerns are solved. Thanks!

---

> > > ### Author Response · Authors · 2025-08-08
> > > **Thanks for your thoughtful comment**
> > >
> > > ### **1 Reconstruction uncertainty undermines AR’s applicability.**
> > >
> > > The aerospace use cases are not just anecdotal. We did not provide enough context for the applications (pages 8-9) because we centered the paper around the core of AR. However, the results presented in Tables 2-3 and Figures 4-5 are from the analysis of a classifier and a pose estimator from the formation flight application; the airport scenario in Table 1 corresponds to the automated landing task. Visualizations of both scenes are provided in Table 6 (see updated_appendix.pdf in the supplementary materials).
> > >
> > > —-
> > >
> > > **Additional results**
> > >
> > > We have performed additional experiments for the formation flight use case. Recall, the perception pipeline is: camera mounted on one airplane→ image→ classifier (ResNet-9) or a relative pose estimator (GateNet). We have used AR to analyze the performance of the classifier and the pose estimator in two scenarios:
> > >
> > > S1: In a fixed 3DGS scene the camera pose varies in a 3D cone defined by relative distance in [90,150] ft, relative elevation angle $[13^\circ,30^\circ]$, and relative yaw angle $[−12.2^\circ,12.2^\circ]$.
> > >
> > > S2: The airplane in foreground is reconstructed using 3DGS, and the background is generated by pixel-wise interpolation between a clowdy sky and a mountainous landscape (visually approximates levels of cloud cover). This scene is verified over the relative distance range of [90,150]ft and interpolation coefficient [0,1] (0 = sky, 1 = mountain).
> > >
> > > In both, the objective is to verify the classifier and the pose estimation (within 2 ft of the ground-truth) or identify regions where they fail.
> > >
> > > *Results*.
> > >
> > >   - Out of 1,000 partitions of the 3D pose space in S1,
> > >
> > >     - Classifier: verified: 930, inconclusive: 8, failed: 62.
> > >
> > >     - Pose estimator: verified: 688, inconclusive: 2, failed: 310.
> > >
> > >   - Out of 100 partitions of the 2D space in S2,
> > >
> > >     - Classifier: verified: 98, failed: 2.
> > >
> > >     - Pose estimator: verified: 19, failed: 81.
> > >
> > > How are the results of verification used? As mentioned earlier, this type of analysis (even with modest scene variations in S2) is considered an advancement and charts a path towards certified visual autonomy. Down the road, as certification standards for autonomous systems become established, the certification artifacts (e.g., the models, the regions, and results) could be used by an organization like the FAA as evidence for trusting an autonomous system (just like currently test suites and model checking results are accepted as part of the DO178C Airborne systems certification standard).
> > >
> > > In addition, a *failed* region guarantees incorrect or unsafe outputs and can be used for improving the perception models (e.g., via data augmentation). The inconclusive regions should call for further investigation during the design and certification processes.
> > >
> > > —--
> > >
> > > ### **2-3 Experiments restricted to simplified setups, insufficient real-world data, and soundness of AR, and other uncertainties in the rendering process**
> > >
> > > We are not using empirical sampling to support the soundness of AR. Soundness is based on linear relational approximation (Theorem 1). We are providing empirical sampling as a visual sanity check.
> > >
> > > Our experiments demonstrate that AR scales to 3DGS scenes with over 500k Gaussians (e.g., Garden and Airport; see appendix), and downstream analysis (with CROWN) supports neural networks as large as VGGNet-16.
> > >
> > > In pose-perturbed classification and pose estimation tasks, low-density sampling often misses violations AR detects, while high-density sampling finds violations but requires significant time. In S1 (discussed above), sampling 10 points per subregion finds 253 of 310 true failures; 100 points find 306; and 1,000 points find all, taking 8 minutes—which is about 15% of AR’s runtime. AR offers a more rigorous option, albeit with longer verification times (recall Q2 W2 discussion).
> > >
> > > We acknowledge that our current formulation assumes a pinhole camera model and adopts standard rendering procedures such as rasterization (3DGS) and ray casting (NeRF). These assumptions, while simplified, are widely adopted for Sim2Real transfer in robotics and serve as the foundation for many modern variants. Providing formal guarantees under these standard settings already represents a significant step forward.
> > >
> > > Incorporating reconstruction uncertainty—e.g., camera calibration errors—is an important direction for future work to strengthen the connection between formal guarantees, debugging, and real-world deployments. Ultimately, “All models are wrong, but some are useful”—and 3DGS reconstructions (even without EVER and 3DGUT) are accepted as useful approximations for Sim2Real transfer.
> > >
> > > We reemphasize that AR is the first framework to provide formal guarantees for the learnable scene-based rendering process, and to our best knowledge, our work stands at the forefront of end-to-end formal verification for vision pipelines involving scene rendering and downstream tasks.

---

### Comment · Area_Chair_MfAm · 2025-08-01
**Author-Reviewer Discussion Period (July 31 - Aug 6)**

The author rebuttals are now posted.

To reviewers:
Please carefully read the *all* reviews and author responses, and engage in an open exchange with the authors.
Please post the response to the authors as soon as possible, so that we can have enough time for back-and-forth discussion with the authors.

---

> ### Comment · Area_Chair_MfAm · 2025-08-05
> **Discussion Period Ends Soon (Aug 6)!**
>
> Dear reviewers,
> Thanks so much for reviewing the paper. The discussion period ends soon. To ensure enough time to discuss this with the authors, please actively engage in the discussions with them if you have not done so.

---

### Decision · Program_Chairs · 2025-09-17

**Decision:**

Accept (spotlight)

**Comment:**

This paper proposes an abstract rendering, an approximate method to estimate the lower and upper bounds of the rendered images from a 3D scene, given over a continuous range of camera poses. The paper presents algorithms for propagating input uncertainty through the rendering pipeline, constructing conservative bounds on the output images for verification.

The reviewers highly value the novelty of this work. In particular, the paper discusses the *abstract* rendering using noisy camera poses (bounded by some range) in Gaussian splatting (GS) and NeRF. The method would also be interesting from the viewpoint of Bayesian optimization.

During the reviewer discussions, all authors' opinions converged on the acceptance of the paper.